

# Calibration of a large-scale hydrological model using satellite-based soil moisture and evapotranspiration products

Patricia Lopez Lopez[1,2], Edwin Sutanudjaja[2], Jaap Schellekens[1], Geert Sterk[2], and Marc Bierkens[2,3]

[1]Deltares, Delft, the Netherlands
[2]Department of Physical Geography, Faculty of Geosciences, Utrecht University, Utrecht, the Netherlands
[3]Deltares, Utrecht, the Netherlands

*Correspondence to:* Patricia López López (patricia.lopez@deltares.nl)

**Abstract.**

A considerable number of river basins around the world lack sufficient ground observations of hydro-meteorological data for effective water resources assessment and management. Several approaches can be developed to increase the quality and availability of data in these poorly gauged or ungauged river basins, and among those, the use of earth observations products

has recently become promising. Earth observations of various environmental variables can be used potentially to increase the knowledge about the hydrological processes in the basin and to improve streamflow model estimates, via assimilation or calibration. The present study aims to calibrate the large-scale hydrological model PCR-GLOBWB using satellite-based products of evapotranspiration and soil moisture for the Moroccan Oum Er Rbia basin. Daily simulations at a spatial resolution of 5 arcmin x 5 arcmin are performed with varying parameters values for the 32-year period 1979-2010. Five different calibration

scenarios are inter-compared: (i) reference scenario using the hydrological model with the standard parameterization, ii) calibration using in-situ observed discharge time series, (iii) calibration using GLEAM actual evapotranspiration time series, (iv) calibration using ESA CCI surface soil moisture time series and (v) step-wise calibration using GLEAM actual evapotranspiration and ESA CCI surface soil moisture time series. The impact on discharge estimates of precipitation in comparison with model parameters calibration is investigated using three global precipitation products, including EI, WFDEI and MSWEP.

Results show that GLEAM evapotranspiration and ESA CCI soil moisture may be used for model calibration resulting in reasonable discharge estimates (NSE values from 0.5 to 0.75), although better model performance is achieved when the model is calibrated with in-situ streamflow observations. Independent calibration based on only evapotranspiration or soil moisture observations improves model predictions to a lesser extent. Precipitation input affects to discharge estimates more than calibrating model parameters. The use of WFDEI precipitation leads to the lowest model performances. Apart from the in-situ

discharge calibration scenario, the highest discharge improvement is obtained when EI and MSWEP precipitation products are used in combination with a step-wise calibration approach based on evapotranspiration and soil moisture observations. This study opens up the possibility to use globally available earth observations and reanalysis products of precipitation, evapotranspiration and soil moisture into large-scale hydrological models to estimate discharge at a river basin scale.



## 1 Introduction

To assess and manage the available water resources within a river basin, good estimates of hydro-meteorological data, such as precipitation, temperature and streamflow, are required. Yet many river basins around the world still have a limited number of in-situ observations. Consequently, most of the watersheds in the world are either ungauged or poorly gauged (Loukas and Vasiliades, 2014). This refers to basins where streamflow or meteorological data are not measured or measurements are discontinued due to termination of a measurement project and/or instrument failure (Sivapalan et al., 2003). Developing novel strategies to enhance and improve available datasets and hydrological models have turned into a key issue in the so-called "ungauged basins" (Hrachowitz et al., 2013).

To overcome the lack of hydro-meteorological data, a promising approach is the use of the recently developed global earth observations and reanalysis products to increase the quality and availability of a wide variety of environmental data. In the last decades, radar and satellite technologies have improved and have become more broadly available providing diverse hydro-meteorological datasets at finer spatial and temporal resolutions: precipitation -CMORPH (Joyce et al., 2004); TRMM (Huffman et al., 2007); etc.-, soil moisture -AMSR-E (Njoku et al., 2003); ESA-CCI (Dorigo et al., 2015); etc.-, total water storage -GRACE (Tapley et al., 2004); etc.-, evapotranspiration -SEBAL (Bastiaanssen et al., 1998); MODIS (Mu et al., 2011); GLEAM (Miralles et al., 2011b); etc.-, etc. Previous studies have demonstrated the possibility of using these global datasets to better understand the hydrological processes in a river catchment (Kite and Droogers, 2000; Vereecken et al., 2008; Seneviratne et al., 2010; Hafeez et al., 2011). Moreover, some studies have proven the potential of earth observations products to improve streamflow model estimates in ungauged basins, through assimilation and calibration techniques. The assimilation of remotely sensed data, including, for example, soil moisture (Parajka et al., 2006; Brocca et al., 2012; López López et al., 2016) or snow cover (Roy et al., 2010; Thirel et al., 2013), may positively impact discharge predictions. Furthermore, calibration to hydro-meteorological variables obtained through satellite and radar can enhance traditional calibration methodologies based on streamflow observations (Immerzeel and Droogers, 2008; Immerzeel et al., 2009; Winsemius et al., 2009).

It has been recognized (Fenicia et al., 2007; Gupta et al., 2008) that because hydrological models are traditionally calibrated based on the comparison between observed and modeled hydrographs only, and often for a limited number of locations, problems of over-parameterization occur, i. e. similar model results are obtained with more than one parameters combination. Different approaches can be followed to overcome this problem: (i) calibrating streamflow to multiple objective functions -multiobjective calibration- (Gupta et al., 1998; Khu and Madsen, 2005) and/or (ii) calibrating to multiple variables, not only to streamflow -step-wise calibration- (Immerzeel and Droogers, 2008; Wanders et al., 2014). Fenicia et al. (2007) applied and compared both calibration approaches in two models of different levels of complexity, using a multi-step calibration approach to identify the parameters according to multiple objective functions for high and low flows. In ungauged basins, where none or limited in-situ meteorological and streamflow observations can be found, the second approach may provide a suitable solution using globally available remotely sensed data. Additionally, this calibration approach is not limited to fitting model estimates to discharge data, but it helps to a better understanding of the processes happening within the catchment.





Several studies in the scientific literature investigated the calibration of model parameters to environmental variables different to streamflow. Campo et al. (2006) used soil moisture information from radar images from ERS-2 sensors to parameterize the hydrological model MOBIDIC. Immerzeel and Droogers (2008) calibrated the hydrological model SWAT based on satellite evapotranspiration from MODIS satellite images. Lo et al. (2010) improved the parameter estimation of the Community Land

Model 3.0. using GRACE total water storage data. In Rientjes et al. (2013), the HBV model was calibrated on satellite based evapotranspiration from MODIS and streamflow. Wanders et al. (2014) calibrated model parameters of LISFLOOD based on discharge and soil moisture observations acquired by AMSR-E, SMOS and ASCAT. Sutanudjaja et al. (2014) calibrated the large-extent model PCR-GLOBWB using streamflow and soil water index information derived from the ERS scatterometers. Isenstein et al. (2015) calibrated the VIC hydrological model using snow covered area from MODIS satellite data. At a global

scale, Beck et al. (2016b) used parameter regionalization to calibrate a HBV model. Furthermore, it was shown that a precipitation product based on a merger of reanalysis, earth observations and gauge information provided best discharge estimates (Beck et al., 2016a). In most of these studies, model calibration was based on only one system variable different to streamflow, such as soil moisture, total water storage, evapotranspiration or snow, and some of them combined one of those observations with discharge data. Various environmental variables, independently and in combination with streamflow observations, may be used

with calibration objectives based on the hydrological processes that each model component and parameter represents inside the overall system. In the present study, this is tested by comparing multiple calibration scenarios based on evapotranspiration, soil moisture and discharge data.

In 2016, a new enhanced version of GLEAM evapotranspiration (Martens et al., 2016b) was released and could potentially be used to improve hydrological model estimates, leading to a better understanding of hydrological processes. Furthermore,

soil moisture data previously used for calibration were obtained from AMSR-E, SMOS and ASCAT instruments at a spatial resolution of 35-50 km. Recently a new satellite soil moisture product from ESA CCI (Dorigo et al., 2015) was developed at a higher quality and a finer spatial resolution of approximately 25 km, combining the benefits of satellite active microwave data and passive satellite microwave radiometry. The present work investigates whether the use of these new satellite-based data products for model calibration could positively impact discharge estimates.

All previously mentioned calibration experiments were performed for well known river basins, such as the Rhine-Meuse river basin, with a good coverage of in-situ hydro-meteorological data. In the present work, the study area is the Oum Er Rbia river basin located in Morocco, where ground observations are spatially sparse and scarce in number. Further insight into model calibration procedures for poorly gauged or ungauged basins could be gained from this study.

When calibrating a hydrological model, meteorological forcing data have a significant role in the accuracy of model pre-

dictions. Oudin et al. (2006) show that precipitation errors significantly decrease model performance, indicating that model parameters may be wrongly calibrated to adjust for those errors in precipitation data. Looper et al. (2012) assessed the relative importance of precipitation and calibration for the accuracy of streamflow predictions, concluding that precipitation may be more important to improve model predictions than calibrating model parameters, depending on the characteristics of the basin, the structure of the hydrological model, the time period analysed and the precipitation product used. In the present study, the

influence of precipitation is considered and three different global precipitation products are used and inter-compared.





This study aims to calibrate a large-scale hydrological model using soil moisture and evapotranspiration observations alone and to compare its discharge estimates to those obtained when the model is traditionally calibrated to streamflow data. To this end, five different calibration approaches are tried by using different calibration strategies with several variables, including streamflow, soil moisture and evapotranspiration. The calibration scenarios are (i) reference scenario using the hydrological

model with the standard parameterization, (ii) calibration using in-situ observed discharge time series, (iii) calibration using GLEAM actual evapotranspiration time series, (iv) calibration using ESA CCI surface soil moisture time series and (v) step-wise calibration using GLEAM actual evapotranspiration and ESA CCI surface soil moisture time series. A priori, it is expected that calibrating to streamflow observations yields to the best discharge estimates, and that the step-wise calibration using soil moisture and evapotranspiration provides better results than the calibration scenarios based only on soil moisture or

evapotranspiration.

The large-scale hydrological model used for the study is the new version of PCR-GLOBWB (PCR-GLOBWB 2.0, https://github.com/UU-Hydro/PCR-GLOBWB_model, Sutanudjaja et al., 2016) and it is forced with three different global precipitation products, which are initially analysed and compared. In this context, it is interesting to investigate whether the predictions of large-scale hydrological models driven with global meteorological data can be improved when calibrated to

globally available observations different to streamflow, such as satellite evapotranspiration and soil moisture. Understanding the potential gain of calibrating large-scale models to remotely sensed observations may have benefits for water resources management, especially in data-poor river basins. The Moroccan Oum Er Rbia basin was chosen as a study area as an example of a river basin with limited number of gauging and weather stations, where the recent land use changes, the increased industrial activity and the impacts of climate change have led to water shortage from surface and groundwater resources. Therefore,

developing new strategies to model this watershed is highly relevant to improve water management and assessment of the water availability within the basin.

The novel aspects and new contributions of this work include the use and comparison of three different and recently generated global precipitation products, the exploration of calibration techniques based on earth observations of soil moisture and evapotranspiration and their application into a large-scale hydrological model to provide streamflow estimates in the ungauged

river basin of Oum Er Rbia in Morocco.

## 2 Study area

The study area is the Oum Er Rbia River basin, which is situated in the central-west region of Morocco between the Atlas Mountains to the south and the Mesetian area to the north flowing into the Atlantic Ocean (Fig. 1). The basin's topography ranges from 2,800 m in the southern upstream zone to 150 m in the northern downstream zone. The Oum Er Rbia is the second

largest river in Morocco with a total length of 550 km and it drains an area of approximately 38,025 km$^2$.

The climate in the coastal and mountainous areas is Mediterranean, characterized with high temperatures in the summer and warm autumn and winter months with rainfall, and semi-arid in the central plain (Jones et al., 2013). Precipitation increases





from downstream to upstream areas in the mountains. The mean annual precipitation and temperature are 400 mm and 18°C, respectively. Approximately 70 % to 80 % of the annual rainfall is concentrated in the period from October to May.

The lowlands of the basin are mainly covered with rain-fed and irrigated agriculture fields and the upstream regions are a combination of Mediterranean forests, woodlands and scrubs. The geology of the area is mostly composed of limestone, marls

and sandstone with a karst aquifer in the Atlas Mountains and a multi-layered system of superficial and deep aquifers in the western plains (Bouchaou et al., 2009).

## 3   Methodology

### 3.1   Large-scale hydrological model: PCR-GLOBWB

The large-scale hydrological model PCR-GLOBWB 2.0 (https://github.com/UU-Hydro/PCR-GLOBWB_model, Sutanudjaja

et al., 2016) was used at a spatial resolution of 5 arcmin x 5 arcmin (approximately 10 km x 10 km at the equator) and at a daily temporal resolution. PCR-GLOBWB is a leaky-bucket type of model applied on a cell-by-cell basis. Figure 2 illustrates a schematic representation of the structure of PCR-GLOBWB model. For each grid cell and time step, the model determines the water balance considering the following water storage components: soil moisture, groundwater, surface water, interception storage and snow. The soil is divided into three vertical layers representing the top 5 cm of soil (depth $Z_1 \leq 5$ cm), the following

25 cm of soil (depth $Z_2 \leq 30$ cm) and the remaining 120 cm of soil (depth $Z_3 \leq 150$ cm), in which the storages are symbolized as $S_1$, $S_2$ and $S_3$, respectively. The underlying groundwater store ($S_4$) consists of two layers: an active or renewable layer and a non-active or non-renewable layer of fossil water, in which the storages are symbolized as $S_{4act}$ and $S_{4fos}$, respectively. The model also includes the water exchange processes between the top layer and the atmosphere (precipitation, evapotranspiration and snowmelt), between the soil layers (percolation and capillary rise) and between the soil layers and the active layer of the

groundwater store (groundwater recharge, discharge to baseflow and capillary rise). Each grid cell is divided into sub-grids considering variations of elevation, vegetation, soil and land cover. Five land cover types are distinguished: irrigated paddy field, irrigated non-paddy field, grassland (short natural vegetation), forest (tall natural vegetation) and open water. To compute the total runoff of every grid cell, the model includes direct runoff ($Q_{DR}$), shallow sub-surface flow from the third soil layer ($Q_{SF}$), and baseflow from the active groundwater layer ($Q_{BF}$). The total runoff is accumulated from all grid cells and routed

along the drainage network to obtain the river discharge ($Q_{channel}$). The PCR-GLOBWB model version used here (Sutanudjaja et al., 2016) simulates water availability and water abstraction, including reservoirs and domestic, industrial, livestock and irrigational water demands. The following subsections briefly describe the model components and the parameters relevant for the present calibration study. The reader is referred to Sutanudjaja et al. (2011) and Sutanudjaja et al. (2014) for a more detailed explanation.





### 3.1.1 Direct or surface runoff

The amount of water that goes into the soil is the net precipitation ($P_n$) resulting from the surplus of precipitation above the interception capacity and the excess melt water from the snow pack. $P_n$ is partitioned into direct runoff ($Q_{DR}$) and net infiltration to the first soil layer ($P_{01}$). The partitioning is done using the Improved Arno Scheme (Hagemann and Gates,

2003), in which the fraction of saturated soil of a cell is estimated based on the cell-minimum capacity ($W_{min}$), the cell-average actual storage ($W_{act}=S_1+S_2+S_3$) and the water capacity for the entire soil profile ($W_{max}= SC_1+SC_2+SC_3$, $SC_n$: soil water capacity for layer $n$). If $W_{min} = 0$, direct runoff always occurs for a rainfall event. If $W_{min} > 0$, an event $P_n$ only generates runoff $Q_{DR}$ if $W_{act} > W_{min}$. $W_{min}$ is therefore an important parameter which governs runoff generation response time.

### 3.1.2 Vertical water exchanges between soil and groundwater stores and shallow sub-surface flow

Net infiltration water into the first soil layer ($P_{01}$) is transferred through the remaining soil layers. Vertical water exchanges occur between the first and the second layers ($P_{12}$), between the second and the third layers ($P_{23}$) and between the third soil layer and the active layer of the groundwater store ($P_{34}$). $P_{12}$,$P_{23}$ and $P_{34}$ consist of downward percolation and upward capillary rise, which depend on the degree of saturation ($s_1 = S_1/SC_1$ ,$s_2 = S_2/SC_2$ and $s_3 = S_3/SC_3$ ) and the unsaturated hydraulic conductivity of each soil layer ($K_{sat\,1}$,$K_{sat\,2}$ and $K_{sat\,3}$). If $s_1$>$s_2$ , percolation is equal to $K_{sat\,1}$; whereas if $s_2$>$s_1$,

capillary rise is equal to $K_{sat\,2} \times (1-s_1)$, being $(1-s_1)$ the moisture deficit in the first soil layer. $K_{sat\,1}$, $K_{sat\,2}$ and $K_{sat\,3}$ controls the vertical fluxes between the soil layers and the groundwater store which affect significantly to the ground water recharge. Moreover, $K_{sat\,3}$ influences the shallow sub-surface flow from the third soil layer ($Q_{IF}$).

### 3.1.3 Baseflow

The last component that contributes to the total runoff for each grid cell is the baseflow from the active groundwater layer ($Q_{BF}$).

$Q_{BF}$ is calculated as $Q_{BF} = S_{4act} \times J$, where $J$ is the baseflow recession coefficient and depends on the aquifer transmissivity and the aquifer specific yield. Therefore, $J$ controls the direct contribution of groundwater store to the total runoff and hence, to the river discharge.

### 3.1.4 Evapotranspiration

Actual evapotranspiration consist of transpiration ($E_t$), bare soil evaporation from the top soil layer ($E_b$), open-water evapo-

ration ($E_w$), interception loss ($E_i$) and evaporation from the melt water store in the snow pack ($E_s$). Each evapotranspiration component is calculated using the reference potential evapotranspiration ($E_{(p,0)}$) as basis and the corresponding factor coefficients related with vegetation cover fraction, crop and land cover type, surface water bodies, water stress and the interception flux.





### 3.2 Data

#### 3.2.1 Meteorological data

The meteorological data required to force PCR-GLOBWB are air temperature, precipitation and reference potential evapotranspiration. In principle, in-situ precipitation and air temperature measurements could be obtained from the climate datasets developed by the World Meteorological Organization (WMO) Regional Climate Centers (http://www.wmo.int/pages/themes/climate/).

A total of 5 weather stations were found in the Oum Er Rbia basin (2) and its surrounding area (3). These measurements were too scarce in number and spatially sparse to cover the entire basin. Therefore, air temperature and precipitation were obtained from the WATCH Forcing Data methodology applied to ERA-Interim reanalysis data (WFDEI) at an original spatial resolution of $0.5^o$ x $0.5^o$ (Weedon et al., 2014). Precipitation and air temperature were downscaled from the original spatial resolution to a $0.08^o$ x $0.08^o$ grid using precipitation and temperature lapse rates derived from the 10' CRU-CL2.0 data

(New et al., 2002) through a linear regression analysis (Sutanudjaja et al., 2011). Reference potential evapotranspiration was obtained through the FAO Penman-Monteith equation. Reference potential evapotranspiration was downscaled from the original spatial resolution to a $0.08^o$ x $0.08^o$ grid using the e2o-downscaling-tools generated within the European Union Seventh Framework Programme (FP7/2007-2013) Global Earth Observation for integrated water resource assessment: eartH2Observe

(https://github.com/earth2observe/downscaling-tools).

To test model sensitivity to precipitation, air temperature and reference potential evapotranspiration were fixed and two additional global precipitation products were used: (i) ERA-Interim reanalysis data (EI) from the European Centre for Medium-range Weather Forecasts (ECMWF) at the original spatial resolution of $0.5^o$ x $0.5^o$ (Dee et al., 2011) and (ii) Multi-Source Weighted-Ensemble Precipitation data (MSWEP) by merging gauge, satellite and reanalysis data at the original spatial resolu-

tion of $0.25^o$ x $0.25^o$ (Beck et al., 2016c).

To inter-compare the precipitation products, the annual mean precipitation for the study time period (1979-2010) for each forcing dataset was calculated (Figs. 3a, 3b and 3c). In addition to the spatial resolution difference, MSWEP is able to capture the rainfall pattern over the Atlas Mountains rather well, which is only roughly distinguished by WFDEI and unrecognized by EI. The finer spatial resolution and the combination of reanalysis, satellite and in-situ data are probably the reasons for its more

plausible spatial pattern. Furthermore, inter-annual variability of precipitation products was analyzed (Fig 3d). WFDEI ranges from 4.5 mm in July to 57 mm in February, whereas EI and MSWEP show a lesser variability with precipitation values from 10.5 mm in July to 42.6 mm in November. Smaller differences between WFDEI and EI and MSWEP are observed during the summer months. EI and MSWEP show similar temporal precipitation patterns. Annual mean precipitation over the entire basin obtained with MSWEP (355.15 mm) is approximately 80 mm higher than with EI (276.67 mm). Similar annual median values

are obtained with the three global precipitation products, although the distribution of WFDEI highly differs from the other two products.

Moreover, the global precipitation values were interpolated to the two weather station locations inside the Oum Er Rbia basin, Beni Mellal and Kasba Tadla (Fig. 1), and Nash-Sutcliffe efficiency (NSE), Kling-Gupta efficiency (KGE), Pearson's correlation coefficient (r), Mean Absolute Error (MAE), Root Mean Squared Error (RMSE) and Percent bias (Pbias) between





the interpolated and in-situ ground data were calculated and shown in Fig. 4. A description of the performance metrics with their mathematical formulation is included in section 3.4. Overall, EI and MSWEP provided a better fit to the station data compared to WFDEI, with higher NSE ($\sim 0.54$ for EI / $\sim 0.54$ for MSWEP), KGE ($\sim 0.60$ for EI / $\sim 0.54$ for MSWEP) and r ($\sim 0.74$ for EI / $\sim 0.85$ for MSWEP) and a MAE ($\sim 0.52$ mm for EI / $\sim 0.46$ mm for MSWEP) and a RMSE ($\sim 0.80$ mm for
EI / $\sim 0.80$ mm for MSWEP) approximately 0.25 - 0.40 mm lower than the other product. EI shows the lowest Pbias at both weather stations, with a value of less than 10 %.

### 3.2.2 Discharge data

Daily river gauge data were obtained from the Oum Er Rbia Hydraulic Agency (ABHOER). Gauge measurements from two gauges in the western region of the basin were used in this study (Fig. 1): Ait Ouchene and Mechra Eddahk. Table 1 summarizes
some key hydrological data.

### 3.2.3 Evapotranspiration data

The GLEAM (Global Land Evaporation Amsterdam Model - http://www.gleam.eu/ - ) evapotranspiration product version 3.0a (GLEAM_v3.0a), generated by VU Amsterdam in collaboration with Ghent University (Miralles et al., 2011b;Miralles et al., 2011a; Martens et al., 2016b), was used to calibrate PCR-GLOBWB. The product consists of a global dataset based
on reanalysis net radiation and air temperature, satellite and gauged-based precipitation, Vegetation Optical Depth (VOD) and snow water equivalents spanning the 35-year period 1980-2014. GLEAM separately estimates the different components of terrestrial evaporation, including transpiration, interception loss, bare-soil evaporation, snow sublimation and open-water evaporation. To this end, it consists of four modules: the evaporation module, the stress module, the soil-water balance module and the rainfall interception model (Martens et al., 2016a). GLEAM ($0.25^o$ x $0.25^o$) was interpolated with distance-weighted
average remapping to a $0.08^o$ x $0.08^o$ grid for the period 1980-2010. GLEAM actual evapotranspiration thus obtained was subsequently compared to simulated actual evapotranspiration by PCR-GLOBWB.

### 3.2.4 Soil moisture data

The ESA CCI surface soil moisture combined product version 2.2 (ESA CCI SM v02.2 CP) was generated as part of the European Space Agency (ESA) soil moisture Climate Change Initiative (CCI) project by the Vienna University of Technology
(http://www.esasoilmoisture-cci.org/). A dataset for the 35-year period 1980-2014 of surface soil moisture was produced using C-band scatterometer data (ERS-1/2 AMI scatterometer, MetOp Advanced Scatterometer -ASCAT-) and multi-frequency radiometer data (SMMR, SSM/I, TMI, AMSR-E, Windsat and AMSR2). Soil moisture retrieved using satellite active microwave data and satellite microwave radiometry were merged to make best use of soil moisture data from the different available satellites and sensors (Liu et al., 2011; Liu et al., 2012; Dorigo et al., 2015). ESA CCI surface soil moisture combined product
represents approximately a top soil layer depth of 0.5 - 2 cm. Similarly to GLEAM, ESA CCI SM product at an original spatial





resolution of $0.25^o$ x $0.25^o$ was interpolated with distance-weighted average remapping to $0.08^o$ x $0.08^o$ grid for the period 1980-2010.

ESA CCI surface soil moisture observations were compared to simulated soil moisture with the first of the three vertical soil layers in PCR-GLOBWB. Due to differences in layer depth and/or data characteristics, systematic biases between modelled and observed soil moisture may exist (Reichle and Koster, 2004). To overcome this expected discrepancy and match the remotely sensed observations to the statistics of corresponding hydrological model simulations, a mean-standard deviation ($\mu - \sigma$) matching (Draper et al., 2009) was used. This technique was implemented to rescale simulated soil moisture against ESA CCI surface soil moisture time series to have the same mean and variance.

The adjusted simulated surface soil moisture values $\theta'_{sim}$ were calculated as

$$\theta'_{sim} = \frac{\sigma_{\theta_{obs}}}{\sigma_{\theta_{sim}}} \times (\theta_{sim} - \overline{\theta_{sim}}) + \overline{\theta_{obs}} \tag{1}$$

where $\theta_{\text{sim}}$ are the simulated soil moisture values, $\theta_{\text{obs}}$ is the ESA CCI soil moisture observations, $\sigma_{\theta_{sim}}$ and $\sigma_{\theta_{obs}}$ are the standard deviations of the simulated and observed soil moisture values and $\overline{\theta_{\text{obs}}}$ and $\overline{\theta_{\text{obs}}}$ are the means of the simulated and observed soil moisture values.

### 3.3 Calibration and validation strategy

Alternative single objective calibration approaches based on discharge, actual evapotranspiration and surface soil moisture and a multiobjective calibration approach based on actual evapotranspiration and surface soil moisture were inter-compared. The multiobjective calibration approach consisted in calibrating model parameters in sequential steps to optimize objective functions of multiple variables related with the processes represented by each parameter. These experiments were carried out for five different calibration scenarios: i) reference scenario using the hydrological model with the standard parameterization (S0), ii) calibration using in-situ observed discharge time series (S1), iii) calibration using GLEAM actual evapotranspiration time series (S2), iv) calibration using ESA CCI surface soil moisture time series (S3) and v) step-wise calibration using GLEAM actual evapotranspiration and ESA CCI surface soil moisture time series (S4).

Calibration scenario S0 represents the reference calibration scenario, which was not locally calibrated for the Oum Er Rbia basin, but uses a-priori model parameters derived from vegetation, soil properties and geological information at a global scale (latest model version of PCR-GLOBWB). Calibration scenario S1 aims to calibrate the hydrological model using in-situ discharge observations, following the traditional calibration approach. Calibration scenario S4 represents the multiobjective calibration approach and it consists of a step-wise calibration scheme that attempts to combine the strengths of calibration scenarios S2 and S3. Step one is simply scenario S2, where all the model parameters are allowed to be calibrated based on GLEAM actual evapotranspiration. In step two, those parameters that are clearly identified by calibration scenario S2 are held constant and the remaining parameters are allowed to be calibrated according to ESA CCI surface soil moisture, calibration scenario S3. The five calibration scenarios were analysed for each of the three global precipitation products to study their impact on model parameters calibration and model performance. The calibration scenarios are described in Table 2, including the scenario identifier.





For the calibration using in-situ observed discharge time series (S1), two river gauge observation time series were used (section 3.2.2). The objective function to maximize for the calibration scenarios was Kling-Gupta efficiency (KGE). The mathematical formulation and description of the used objective function are included in section 3.4.

To calibrate PCR-GLOBWB for each of the three precipitation products, 81 runs with different parameter values were simulated: minimum soil water capacity ($W_{\min}$), soil saturated hydraulic conductivites ($K_{\text{sat }1}$, $K_{\text{sat }2}$ and $K_{\text{sat }3}$), baseflow recession coefficient ($J$) and reference potential evapotranspiration ($E_{p,0_{ref}}$). These model parameters, which vary spatially over the basin, influence different model parts of the model behaviour, as it was explained in section 3.1. For the variation of the parameter values, spatially uniform prefactors were used: $f_w$, $f_K$, $f_j$ and $f_e$ (Table 3). The remaining model parameters were kept fixed.

The prefactors to vary model parameter values were referred to the parameters of the S0 calibration scenario. The spatial distribution of the parameters $W_{\min}$, $K_{\text{sat}}$ and $J$ used in S0 scenario can be found in Figure A1 of Appendix A. As reference calibration scenario, S0 prefactors are: $f_w = 1$, $f_K = 0$, $f_j = 1$ and $f_e = 1$. The model performances of all the simulations were evaluated for each of the five calibration scenarios to identify the best parameter sets, based on the obtained prefactors, as the calibrated parameter sets.

All the simulations were performed at a daily temporal resolution for the 32-year period 1979-2010. The 2-year period 1979-1980 was used to spin up the hydrological model until reaching a dynamically steady state. The model was calibrated based on monthly values of discharge, actual evapotranspiration and surface soil moisture. Validation was also carried out at a monthly temporal resolution but exclusively for streamflow, aiming to analyse if similar discharge estimates may be obtained with a calibrated model based on remotely sensed observations (S2, S3 and S4), in comparison with a model traditionally calibrated to in-situ discharge data (S1). The 13-year period 1981-1993 was used for model calibration and during the 17-year period 1994-2010, the model was validated.

## 3.4 Performance metrics

To inter-compare the three global precipitation products six metrics were used: Nash-Sutcliffe efficiency (NSE), Kling-Gupta efficiency (KGE), Pearson's correlation coefficient (r), Mean Absolute Error (MAE), Root Mean Squared Error (RMSE) and Percent bias (Pbias). Moreover, one of those metrics, KGE, was chosen as objective function to calibrate and validate model performance for each calibration scenario. NSE, RMSE and r were also used as additional assessment measurements in the validation procedure.

Nash-Sutcliffe efficiency (Nash and Sutcliffe, 1970), NSE, is defined as

$$NSE = 1 - \frac{\sum_{t=1}^{n}[x(t) - y(t)]^2}{\sum_{t=1}^{n}[y(t) - \overline{y}]^2} \tag{2}$$

where $x(t)$ and $y(t)$ are the modeled and observed variable at $t$ time step (months), $\overline{y}$ is the mean of observed data and $n$ is the total number of observations. NSE is widely used for calibrating and validating hydrological models in terms of discharge. NSE varies from $-\infty$ to 1. If $NSE = 0$, modeled values perform as well as the mean of the observations. If $NSE < 0$, modeled values perform worse than the mean of the observations.




Gupta et al. (2009) analysed various decompositions of NSE and proposed an alternative model performance criteria, Kling-Gupta efficiency (KGE), to avoid the problems that can be derived of using the NSE criterion (e.g. high sensitivity to extreme values and bias). KGE is given as

$$KGE = 1 - \sqrt{(r-1)^2 + (\alpha-1)^2 + (\beta-1)^2} \qquad (3)$$

where $r$ represents the Pearson's correlation coefficient, $\alpha$ is the ratio between the variance of the modeled variable and the variance of the observed variable and $\beta$ is the ratio between the mean of the modeled variable and the mean of the observed variable, i.e. $\beta$ represents the bias. Analogous to NSE, KGE ranges from $-\infty$ to 1 with an ideal value of 1. KGE measures simultaneously bias, variability and correlation.

Pearson's correlation coefficient (Pearson, 1896), r, measures the degree of linear association between modeled and observed values and it is defined as

$$r = \frac{\sum_{t=1}^{n}(x(t)-\overline{x})(y(t)-\overline{y})}{\sqrt{\sum_{t=1}^{n}(x(t)-\overline{x})^2}\sqrt{\sum_{t=1}^{n}(y(t)-\overline{y})^2}} \qquad (4)$$

where $x(t)$ and $y(t)$ are the modeled and observed variable at $t$ time step (months), $\overline{y}$ is the mean of observed data, $\overline{x}$ is the mean of modeled data and $n$ is the total number of observations. r varies within the interval [-1,1]. r is mainly used in hydrological modeling to evaluate the timing of modeled to observed time series.

The following two performance metrics, MAE and RMSE, measure the accuracy of modeled values to observations. Mean Absolute Error, MAE, is calculated as

$$MAE = \frac{1}{n}\sum_{t=1}^{n}(x(t)-y(t)) \qquad (5)$$

and Root Mean Squared Error, RMSE, is given as

$$RMSE = \sqrt{\frac{1}{n}\sum_{t=1}^{n}(x(t)-y(t))^2}. \qquad (6)$$

MAE and RMSE range from 0 to $\infty$ and they are negatively oriented metrics, which means that lower values indicate a better performance. If $MAE = 0$ and $RMSE = 0$, modeled values accurately represent the observations. MAE and RMSE measure the average magnitude of the errors, but MAE is a linear score and RMSE is a quadratic score, which means that the latter one gives relatively high weight to large errors.

Percent bias indicates the average tendency of the modeled values to over- or underestimate the observations. Pbias, is calculated in percentage terms as

$$Pbias = 100 \times \frac{\sum_{t=1}^{n}(x(t)-y(t))}{\sum_{t=1}^{n}y(t)} \qquad (7)$$

The optimal value of Pbias is 0.

## 4 Results

Model parameters were calibrated using discharge, evapotranspiration and soil moisture observations through five different calibration scenarios. To summarize results of all runs obtained for the calibration time period 1981-1993, Figure 5 consists



of four panels corresponding to calibration scenarios S1 - discharge at (a) Ait Ouchene and (b) Mechra Eddahk -, S2 - evapotranspiration - and S3 - soil moisture - . Each of those panels is a matrix of multiple scatterplots, with rows showing the three global precipitation products used as model forcing (EI, WFDEI and MSWEP) and columns showing the prefactors ($f_e$, $f_j$, $f_k$ and $f_w$). Each of the scatterplots show prefactors values on the horizontal axis and discharge performance indicator KGE

on the vertical axis. With this figure, prefactor, and therefore parameter, ranges leading to better and worse performances can be distinguished, as well as their global optimal values. If no optimal value can be inferred, prefactors from the calibration scenario S0 are maintained ($f_e = 1$, $f_j = 0$, $f_k = 0$ and $f_w = 1$).

For calibration scenario S1, scatterplots of Figure 5a and Figure 5b are nearly similar. KGE values were obtained from the comparison between model estimated discharge and observed discharge at Ait Ouchene (Figure 5a) and Mechra Eddahk (Figure

5b). Results show that $f_w$ and $f_e$ are well identified by discharge calibration at both gauging stations when forced with any of the three precipitation products. However, $f_k$ and $f_j$ prefactors show a large spread to estimate the best values from all the runs, hence $f_k = 0$ and $f_j = 0$ are used. For both locations, the optimal discharge performance in terms of KGE is obtained with $f_e = 1.25$ and $f_w = 1.25$. For calibration scenario S2, KGE values were obtained comparing modelled actual evapotranspiration and GLEAM actual evapotranspiration. The scatterplots in Figure 5c show that only prefactor $f_e$ can be clearly identified, whereas

the remainder of the prefactors ($f_j$, $f_w$ and $f_k$) show a large spread, suggesting that evapotranspiration-based calibration may be unreliable in their identification. Therefore, model run with prefactors $f_e = 1.25$, $f_j = 0$, $f_k = 0$ and $f_w = 1$ is considered as the calibrated run based on the evapotranspiration performance.

For calibration scenario S3, KGE values were obtained comparing simulated soil moisture and ESA CCI soil moisture. The scatterplots in Figure 5d show that prefactors $f_w$ and $f_k$ can be identified, $f_w = 1.25$ and $f_k = 0.25$. Prefactors $f_e$ and $f_j$ are

not identifiable when soil moisture is used for calibration. Therefrom, the calibrated run based on soil moisture performance is the model run with prefactors $f_e = 1$, $f_j = 0$, $f_k = 0.25$ and $f_w = 1.25$. This implies that ESA CCI soil moisture may be used to indirectly tune groundwater recharge by calibrating the upper soil saturated hydraulic conductivities, $K_{sat}$.

Calibration scenario S4 attempts to combine the strengths of scenarios S2 and S3. In the first step, all prefactors are allowed to be calibrated to find the highest actual evapotranspiration KGE. In the second step, those prefactors that have been identified

calibrating the model to GLEAM evapotranspiration (i.e. $f_e$) were held constant and the remaining three prefactors are allowed to be calibrated according to ESA CCI soil moisture KGE performance (i.e. $f_w$ and $f_k$). As a result, for calibration scenario S4, the prefactors identified during the evapotranspiration calibration (S2): $f_e = 1.25$ and during the soil moisture calibration (S3): $f_w = 1.25$ and $f_k = 0.25$ are adopted. This step-wise calibration approach using multiple system variables allow to identify more parameters than when those variables are separately used. Nonetheless, prefactor $f_j$ is not clearly identified and its value

for the calibration scenario S0 is used, $f_j = 0$. A combination of high $f_e$, high $f_w$ and high $f_k$ provides the best performance in terms of evapotranspiration and soil moisture.

Once the best runs for each calibration scenario were identified, their discharge performance was checked at the two gauging stations: Ait Ouchene, in Figure 6 and Mechra Eddahk in Figure 7. Each of these figures consists of a matrix of multiple scatteplots, with rows showing the three global precipitation products (EI, WFDEI and MSWEP) and columns showing the

four calibration scenarios (S0, S1, S2, S3 and S4). Each of the scatterplots show observed discharge on the vertical axis and





model estimated discharge on the horizontal axis. The performance indicators NSE and KGE for discharge are included in every scatterplot in Figures 6 and 7.

From Figures 6 and 7, a few general observations can be made. Scatterplots show that estimated discharges are closer to observed discharges at both gauging stations when PCR-GLOBWB is forced with EI precipitation. KGE values for the

reference calibration scenario S0 at Mechra Eddahk are 0.607, 0.325 and 0.561 when EI, WFDEI and MSWEP are used as forcing data respectively. These differences in performance are related to the lower quality of WFDEI in this region compared with the other precipitation products (section 3.2.1).

KGE values at Ait Ouchene station for calibration scenario S0 are lower than for Mechra Eddahk station. This may be due to their different locations within the basin, the former one being situated in the Atlas Mountains, where precipitation estimates

can be less accurate, and in a tributary of the Oum Er Rbia River, whose representation in PCR-GLOBWB can be limited by the model spatial resolution.

Results indicate that the highest discharge performance is obtained when the model is calibrated with in-situ discharge observations (S1). Calibration scenarios based on only evapotranspiration (S2) or soil moisture (S3) also show an increase in KGE values compared with scenario S0, although of a lower magnitude than when ground discharge data is used. Step-

wise calibration scenario S4 based on both evapotranspiration and soil moisture observations leads to further improvements in discharge estimates. For example, when MSWEP precipitation is used to model discharge at Mechra Eddahk station, KGE varies between 0.439, 0.423, 0.369 and 0.328 for calibration scenarios S1, S4, S2 and S3, respectively (KGE = 0.325 for the reference scenario S0). At Ait Ouchene station, KGE varies between 0.520, 0.342, 0.331 and 0.271 for calibration scenarios S1, S4, S2 and S3, respectively (KGE = -0.542 for the reference scenario S0).

The calibrated runs based on evapotranspiration (S2) and soil moisture (S3) observations result in lower discharge performances compared to the reference scenario (S0) at some cases, e.g. when EI precipitation is used at Mechra Eddahk location, KGE (S0) = 0.607, KGE (S2) = 0.534 and KGE (S3) = 0.522. Table 4 summarizes the identifiability of each parameter based on each calibration scenario: in-situ discharge (S1), GLEAM evapotranspiration (S2), ESA CCI soil moisture (S3) and the combination of GLEAM evapotranspiration and ESA CCI soil moisture (S4). The optimal values that were identified are included

in this table, together with the KGE performance values.

Figure 8 and Figure 9 consist of three timeseries graphs, with rows showing the three global precipitation products (EI, WFDEI and MSWEP). In Figure 8, the simulated evapotranspiration timeseries of the reference run (S0) and of the evapotranspiration-calibrated run (S2) are plotted against GLEAM evapotranspiration timeseries (observed) over the entire Oum Er Rbia basin for the validation time period 1994-2011. The performance indicator KGE for evapotranspiration is included in every graph of

Figure 8. In the figure, the calibration procedure based on GLEAM evapotranspiration produces an increase of 27 %, 11 % and 18 % in KGE, when EI, WFDEI and MSWEP precipitation products are respectively used. Similarly to Figure 8, in Figure 9, the modelled soil moisture timeseries of the reference run (S0) and of the soil moisture-calibrated run (S3) are plotted against ESA CCI soil moisture timeseries (observed) over the entire Oum Er Rbia basin for the validation time period 1994-2011. The rescaled soil moisture timeseries (after mean-standard deviation matching technique applied, see section 3.2.4) are shown.

The rescaling technique removes the biases between the simulated and observed soil moisture timeseries. The performance



indicator KGE for soil moisture are given in every plot in Figure 9. From all model runs, the best run with the highest KGE is shown, i.e. 0.859, 0.842 and 0.862 when EI, WFDEI and MSWEP precipitation products are used.

Figure 10 consist of two panels with three discharge timeseries graphs at Ait Ouchene (Figure 10a) and Mechra Eddahk (Figure 10b) stations. Rows show the three global precipitation products (EI, WFDEI and MSWEP). In each graph, the mod-

elled discharge timeseries of the reference run (S0) and of the step-wise calibrated run (S4) are plotted against in-situ discharge timeseries (observed) for the validation time period 1994-2011. The performance indicators for discharge NSE and KGE are included in each graph. The step-wise calibrated run based on GLEAM evapotranspiration and ESA CCI soil moisture (S4) reproduces the observed discharge at the two gauging stations well, as shown in Figure 10 (at monthly temporal resolution), except some simulated extreme peaks which were not observed, e.g. January and June in 2002. This lack of fit may be due to

errors in the precipitation data, because higher discharge differences are shown when WFDEI and MSWEP products are used at both gauging stations.

The evapotranspiration- and soil moisture-calibrated run (S4) improved discharge performance indicators NSE and KGE in comparison with discharge estimates from calibration scenario S0 for most of the cases. For example, NSE increases from -0.604 to 0.380 and KGE increases from 0.011 to 0.617, when MSWEP precipitation is used at Ait Ouchene station. These

increments are lower, but still significant, at Mechra Eddahk station, with NSE values ranging from 0.534 to 0.582 and KGE values varying from 0.648 to 0.710. However, when EI precipitation product is used, the discharge estimates from calibration scenario S4 show a similar or even lower performance than discharge modelled from the reference scenario S0. Thus, an improvement of 7.6 % in NSE and a decrease of 21.8 % in KGE is produced at Ait Ouchene station and a decrease of 1.6 % in NSE and of 9.7 % in KGE is produced at Mechra Eddahk station. This may be due to the quality differences between the

global precipitation products and some errors in other model parameters that were not calibrated, such as the soil thickness parameters.

To further understand the added value of using GLEAM evapotranspiration and ESA CCI soil moisture data for model calibration, the variations of NSE, KGE, RMSE and r between each calibration scenario (S1, S2, S3 and S4) and the reference calibration scenario (S0) were calculated and plotted for the validation time period in Figure 11. This figure consists of multiple

barplots with rows showing the three global precipitation products and columns showing the performance indicators. The variations of the performance metrics are shown with barplots for the two gauging stations. At each location, a positive value of NSE, KGE, RMSE and r means that either S1, S2, S3 or S4 scenario obtained a higher skill score than S0, whereas a negative value means that those scores decreased after calibration. Figure 11 shows that variations of the performance indicators are lower when EI precipitation product is used. The highest differences between the calibration scenarios were obtained when the

model is forced with WFDEI precipitation. This is a consequence of the precipitation discrepancies analysed in section 3.2.1.

In the inter-comparison of the calibration scenarios, calibration scenario using in-situ observed discharge data (S1) obtains overall the highest increase of NSE, KGE and r and the highest reduction of RMSE when any of the precipitation products are used, as it was expected. Similar NSE and KGE increases and RMSE decreases are obtained when the model is calibrated using only soil moisture (S3) and using the combination of evapotranspiration and soil moisture (S4), but larger improvements

in r are obtained with the step-wise calibration scenario (S4). NSE, KGE and r gains when comparing calibration scenarios S2





and S0 are positive, but of a lower magnitude than when model is calibrated in scenarios S3 and S4. The higher performance of scenario S4 may be due to the fact that this calibration approach uses multiple system variables providing more hydrological information and allowing the identification of more parameters than when those variables are separately used.

In each barplot, metrics improvements are larger at Ait Ouchene station than at Mechra Eddahk station. This is due to the
lower discharge performance for the reference calibration scenario S0 at the former gauging location. Note that in some cases where the change in KGE is negative (e.g. when EI precipitation is used at Ait Ouchene station), this is because although there was an improvement in the KGE performance indicator during the calibration time period, when calculating it for the validation time period, it is possible that the metric slightly worsens. Note that some variations in NSE, RMSE and r are small or close to 0, because its calibration was optimised for KGE and not for those particular metrics in terms of discharge.

## 5   Discussion

Results show that GLEAM actual evapotranspiration and ESA CCI soil moisture observations may be used to calibrate determined PCR-GLOBWB model parameters at the ungauged basin of Oum Er Rbia. GLEAM actual evapotranspiration can be used to calibrate the reference potential evapotranspiration ($f_e$) as expected, affecting the water exchange between the top soil layer and the atmosphere and hence the soil water balance. ESA CCI soil moisture data can be used to calibrate the mini-
mum soil water capacity ($f_w$) and the saturated hydraulic conductivities of the soil layers ($f_k$), determining the surface runoff generation response, the shallow sub-surface flow and the groundwater recharge. However, calibration using only GLEAM evapotranspiration data or only ESA CCI soil moisture can result in more than one parameters combination to be optimal in terms of discharge (overparametrization or equifinality problem). To overcome this problem, a step-wise calibration scenario based on both observations, evapotranspiration and soil moisture, is necessary to identify the optimal values of reference poten-
tial evapotranspiration ($f_e$), runoff-infiltration partitioning parameters ($f_w$) and the soil saturated hydraulic conductivity ($f_k$). Nonetheless, neither of these observations can be used to calibrate the baseflow from the active groundwater layer ($f_j$). To identify baseflow recession coefficient parameter ($f_j$) a multiobjective calibration approach to streamflow observations could be followed. Similarly to Fenicia et al. (2007), multiple objective functions may be optimized in sequential steps for high flows, low flows and timing.

A step-wise calibration approach based on GLEAM actual evapotranspiration and ESA CCI soil moisture results in discharge estimates of acceptable accuracy (Moriasi et al., 2007), compared to discharge estimates derived from a model that has been calibrated to in-situ discharge measurements. Results indicate that precipitation impact on streamflow estimates is more significant than the one derived from calibrating model parameters, thus the lower quality of WFDEI compared to EI and MSWEP, decreases model performance and calibration is biased in order to compensate precipitation errors. Further investi-
gation of the effect of precipitation errors on model efficiency, but also on model parameters estimation may be an interesting route for hydrological research (Andréassian et al., 2004; Looper et al., 2012).

This study shows that globally available earth observations, such as evapotranspiration or soil moisture, can be used to further parameterize large-scale hydrological models providing reasonable discharge estimates at regional or basin scale. In





principle, these calibration approaches can be applied and investigated in other basins without or with limited in-situ ground hydro-meteorological data (ungauged basins), not only to estimate discharge, but also to improve the understanding of the hydrological processes in the basin. Results suggest the potential of using other satellite products for hydrological modeling studies, including soil moisture products such as AMSR-E (Njoku et al., 2003) and SMOS (Kerr et al., 2001), evapotranspi-

ration products such as SEBAL (Bastiaanssen et al., 1998) and MOD16 (Nishida, 2003), total water storage products such as GRACE (Tapley et al., 2004), etc. Future studies may investigate step-wise calibration approaches using the combined information from multiple hydrological system variables. By incorporating several data products, different parts or components of the model can be optimized to increase the overall model performance. Alternatively, these hydro-meteorological data which are globally available may be used to identify and develop relationships between different basins using similarities, classification

and scaling frameworks, as presented in previous studies (Samaniego et al., 2010b; Kumar et al., 2013).

Spatially uniform prefactors for the entire Oum Er Rbia basin were used for the variation of the parameter values in this study. Developing novel calibration strategies where prefactors and so, model parameters vary with soil type, land use, elevation and/or other characteristics within the basin would be a promising research route to investigate. Furthermore, the present work inter-compare five calibration scenarios using a brute force method, where several combinations of parameters values are tested

and the best performing is selected. A suggestion for future studies may be to use an Ensemble Kalman Filter to calibrate the hydrological model, as previously presented in literature (Moradkhani et al., 2005; Wanders et al., 2014)

## 6 Summary and conclusions

This study investigates alternative routes to calibrate the large-scale hydrological model PCR-GLOBWB using earth observations globally available for the data-poor river basin of Oum Er Rbia in Morocco. Three global precipitation products, EI,

WFDEI and MSWEP, are inter-compared and applied to force PCR-GLOBWB. Five different calibration scenarios are followed where GLEAM actual evapotranspiration and ESA CCI surface soil moisture data are used to identify model parameters with the aim to improve discharge estimates. In-situ discharge observations are also used for calibration, as they are traditionally used to calibrate hydrological models.

Results show that PCR-GLOBWB may provide reasonable discharge estimates when forced with global precipitation prod-

ucts and calibrated to remotely sensed evapotranspiration and soil moisture observations. Traditional calibration to in-situ discharge measurements results in the highest model performance, as expected. A model calibrated only on evapotranspiration or soil moisture provides reasonable discharge estimates, allowing the identification of those model parameters associated with the hydrological processes that they represent. The step-wise calibration approach using evapotranspiration and soil moisture data combines the benefits of both observations achieving a better discharge performance than when they are separately used.

In the inter-comparison between the three global precipitation products, WFDEI shows the lowest performance, whereas EI and MSWEP perform quite well. Apart from the in-situ discharge calibration scenario, the highest discharge improvement is obtained when the two latter forcing data are used in combination with a step-wise calibration approach based on evapotranspiration and soil moisture observations.





Although there is still room for further research, this study shows that globally available datasets can be used to calibrate a large-scale hydrological models resulting in reasonable discharge estimates for regional-scale catchments and demonstrating that ungauged basins can benefit from further improvements and investments in global observations, global models and their combination.



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



**Table 1.** Hydrological and geographical information of the analysed catchments at the Oum Er Rbia basin.

| Station name | River | Upstream basin area (km$^2$) | Oulet location | | Elevation (m AOD) |
| --- | --- | --- | --- | --- | --- |
| | | | Longitude | Latitude | |
| Ait Ouchene | El Abid | 2350 | -6.180 | 32.225 | 1070 |
| Mechra Eddahk | Oum Er Rbia | 6555 | -6.52 | 32.435 | 406 |



**Table 2.** Calibration scenarios.

| Scenario identifier | Description |
| --- | --- |
| S0 | Reference scenario |
| S1 | Calibration using in-situ observed discharge time series |
| S2 | Calibration using GLEAM actual evapotranspiration times series |
| S3 | Calibration using ESA CCI surface soil moisture time series |
| S4 | Step-wise calibration: using GLEAM actual evapotranspiration and ESA CCI surface soil moisture time series |



**Table 3.** Parameter values used in the calibration processes.

| Parameters ID | Description | Prefactors | Parameter values |
|---|---|---|---|
| $W_{min}$ | Minimum soil water capacity | $f_w \in \{0.75, 1, 1.25\}$ | $W_{min} = f_w * W_{max}$ |
| $K_{sat1}$ | Saturated hydraulic conductivity of $1^{st}$ soil layer | $f_k \in \{-0.25, 0, 0.25\}$ | $log(K_{sat1}) = f_k + log(K_{sat1_{ref}})$ |
| $K_{sat2}$ | Saturated hydraulic conductivity of $2^{nd}$ soil layer | $f_k \in \{-0.25, 0, 0.25\}$ | $log(K_{sat2}) = f_k + log(K_{sat2_{ref}})$ |
| $K_{sat3}$ | Saturated hydraulic conductivity of $3^{rd}$ soil layer | $f_k \in \{-0.25, 0, 0.25\}$ | $log(K_{sat3}) = f_k + log(K_{sat3_{ref}})$ |
| $J$ | Baseflow recession coefficient | $f_j \in \{-0.5, 0, 0.5\}$ | $log(J) = f_j + log(J_{ref})$ |
| $E_{p,0}$ | Reference potential evapotranspiration | $f_e \in \{0.75, 1, 1.25\}$ | $E_{p,0} = f_e * E_{p,0_{ref}}$ |





**Table 4.** Parameter identifiabilities and optimal values for each calibration scenario.

| Calibration scenario | $f_e$ | $f_j$ | $f_k$ | $f_w$ | KGE (Ait Ouchene) | KGE (Mechra Eddahk) |
|---|---|---|---|---|---|---|
| S0 | 1 | 0 | 0 | 1 | 0.470 / -1.906 / -0.542** | 0.607 / 0.325 / 0.561 |
| S1 | 1.25 | NI* | NI | 1.25 | 0.510 / -0.494 / 0.520 | 0.688 / 0.439 / 0.703 |
| S2 | 1.25 | NI | NI | NI | 0.508 / -0.580 / 0.342 | 0.602 / 0.423 / 0.693 |
| S3 | NI | NI | 0.25 | 1.25 | 0.487 / -0.607 / 0.331 | 0.634 / 0.369 / 0.613 |
| S4 | 1.25 | NI | 0.25 | 1.25 | 0.478 / -0.768 / 0.271 | 0.522 / 0.328 / 0.573 |

*NI indicates that the parameter was not identifiable

**KGE values are obtained from observed and simulated discharge when PCR-GLOBWB is
forced with EI / WFDEI / MSWEP

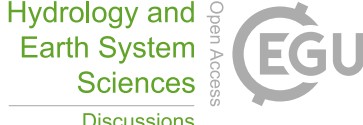

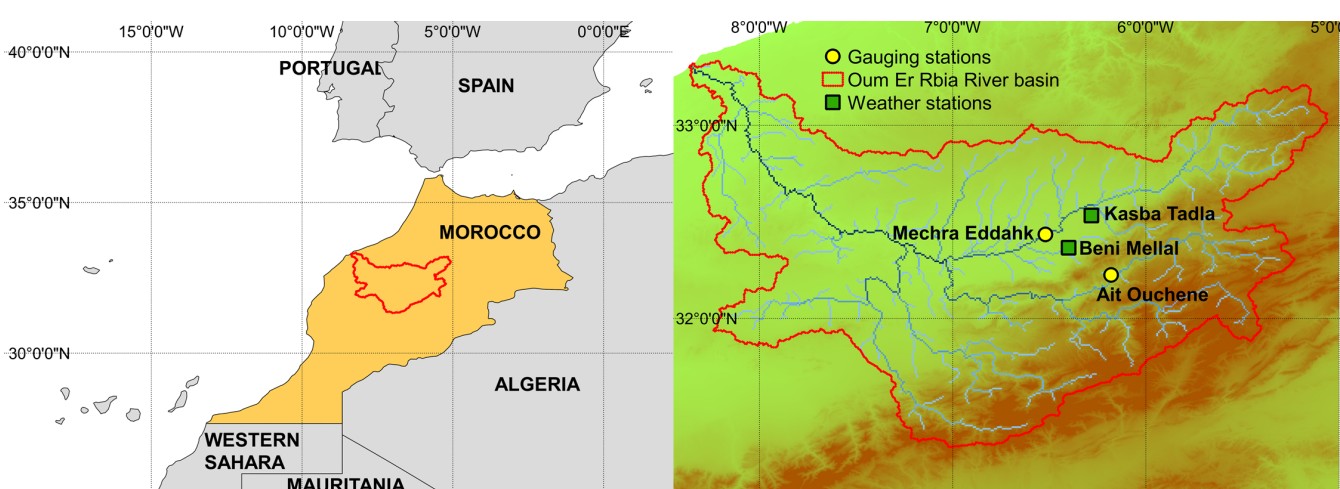

**Figure 1.** Oum Er Rbia River basin and its location in Morocco. Yellow points represent the gauging stations and green squares represent the weather stations.





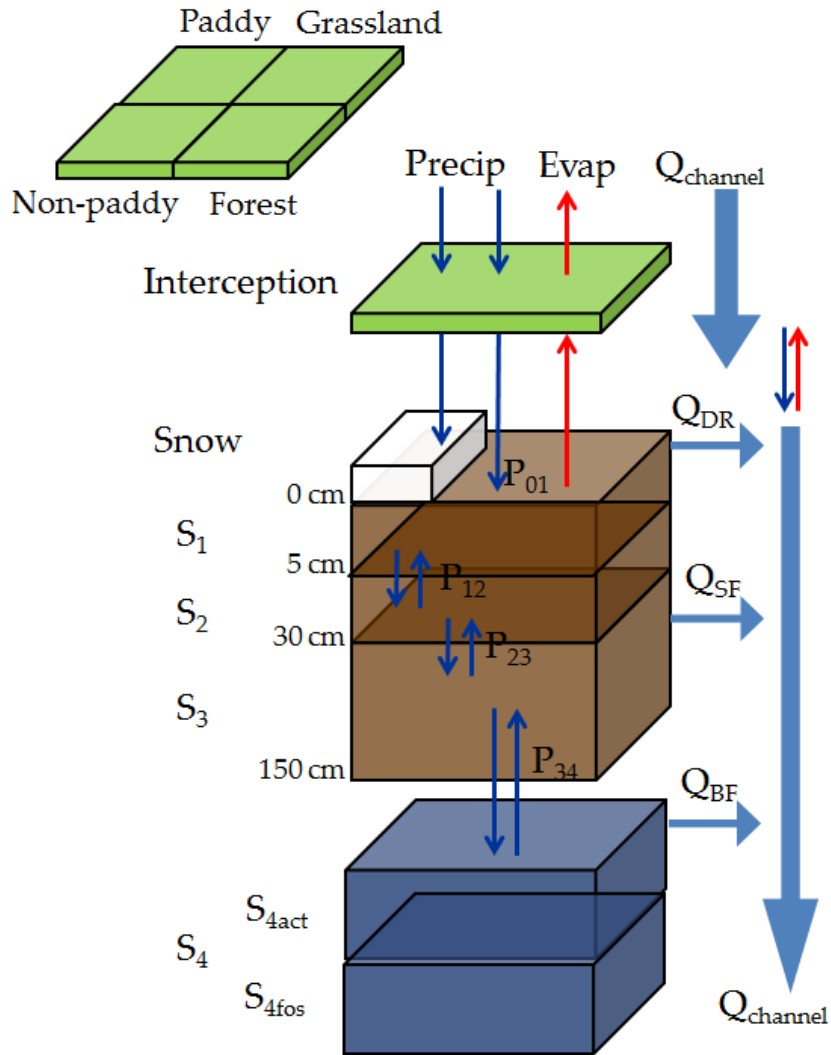

**Figure 2.** PCR-GLOBWB model structure, adapted from Van Beek et al. (2011).





**Figure 3.** (a) EI annual mean precipitation, (b) WFDEI annual mean precipitation and (c) MSWEP annual mean precipitation for 1979-2010 time period and (d) inter-annual variability of EI, WFDEI and MSWEP precipitation products.




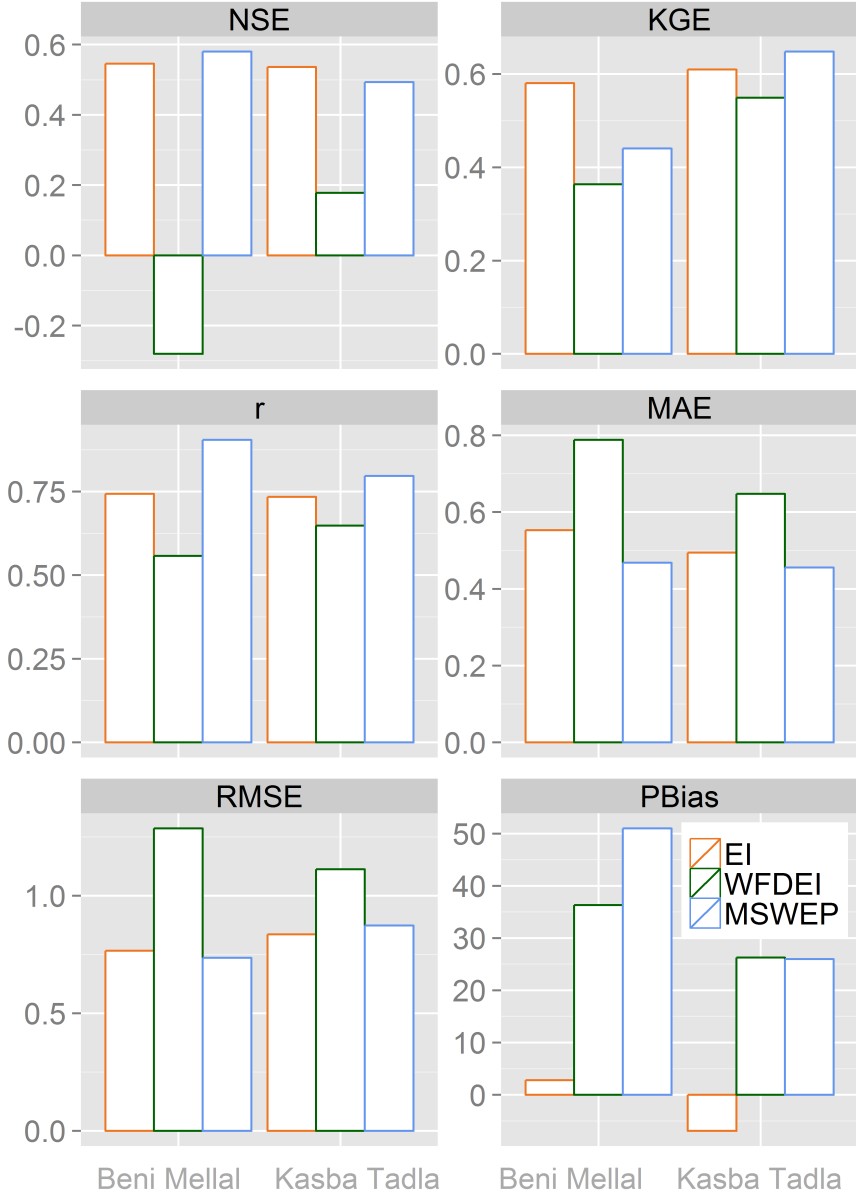

**Figure 4.** Performance metrics of EI, WFDEI and MSWEP precipitation products at Beni Mellal and Kasba Tadla weather stations, including Nash-Sutcliffe efficiency (NSE), Kling-Gupta efficiency (KGE), Pearson's correlation coefficient (r), Mean Absolute Error (MAE), Root Mean Squared Error (RMSE) and Percent bias (Pbias).





**Figure 5.** Scatterplots of discharge performance indicator KGE based on the monthly observations versus prefactors $f_e$, $f_j$, $f_k$ and $f_w$ for the calibration scenarios S1 ((a) Ait Ouchene (b) Mechra Eddahk), S2 (c) and S3 (d). In each panel of the figure, columns show the different calibrated prefactors and rows show the three global precipitation products used as model forcing.





**Figure 6.** Scatterplots of estimated discharge (x-axis) and observed discharge (y-axis) at Ait Ouchene. Rows show the three global precipitation products and columns show the five calibration scenarios.







**Figure 7.** Scatterplots of estimated discharge (x-axis) and observed discharge (y-axis) at Mechra Eddahk. Rows show the three global precipitation products and columns show the five calibration scenarios.



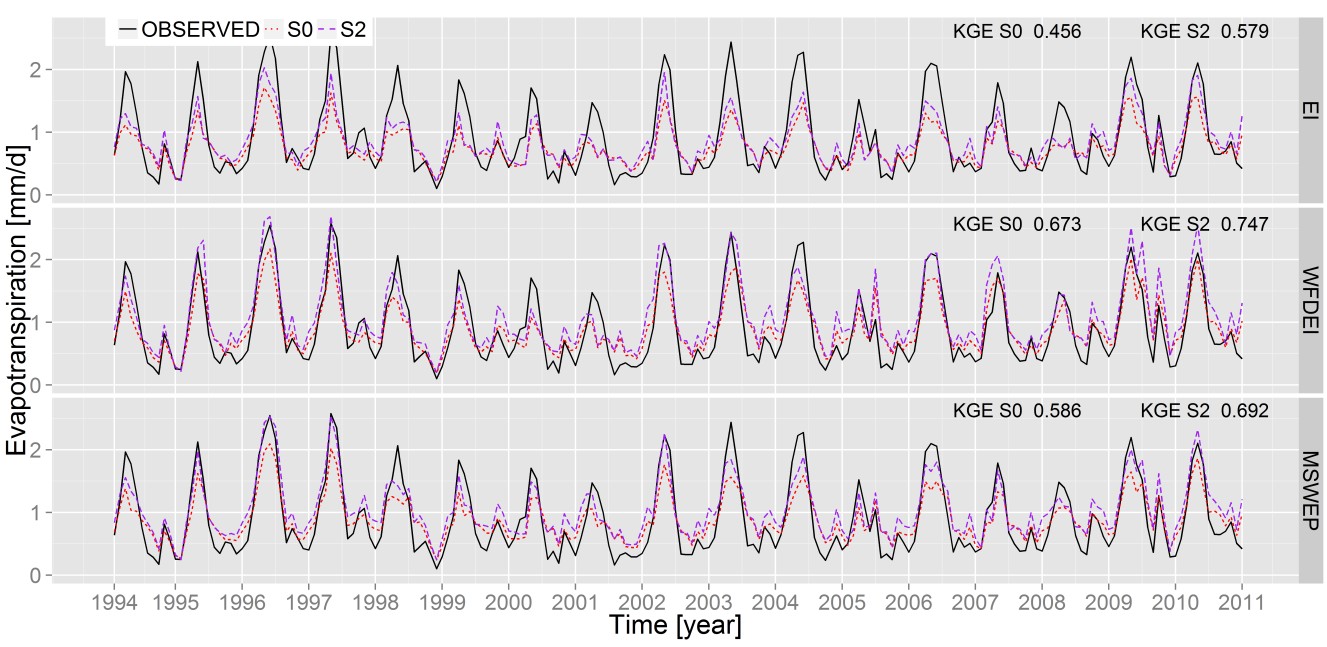

**Figure 8.** Comparisons between GLEAM actual evapotranspiration (black) and estimated actual evapotranspiration (red and purple) time series over the Oum Er Rbia basin for the validation time period. Rows show the three global precipitation products. The red dashed lines represent actual evapotranspiration estimates from calibration scenario S0 (reference scenario) and the purple dashed lines represent the calibrated time series from calibration scenario S2 which are taken from the runs that yield the best simulations.





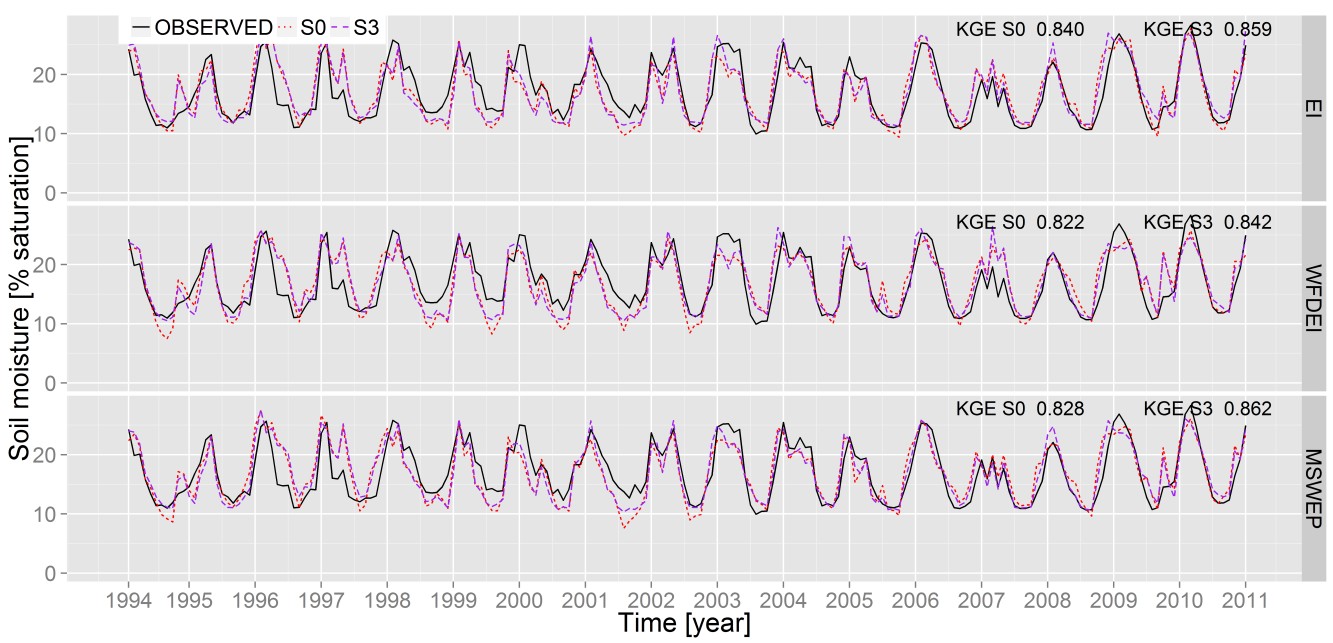

**Figure 9.** Comparisons between ESA CCI soil moisture (black) and estimated soil moisture (red and purple) time series over the Oum Er Rbia basin for the validation time period. Rows show the three global precipitation products. The red dashed lines represent soil moisture estimates from calibration scenario S0 (reference scenario) and the purple dashed lines represent the calibrated time series from calibration scenario S3 which are taken from the runs that yield the best simulations.



**Figure 10.** Comparisons between observed discharge (black) and estimated discharge (red and purple) time series at a) Ait Ouchene and b) Mechra Eddahk. Rows show the three global precipitation products. The red dashed lines represent discharge estimates from calibration scenario S0 (reference scenario) and the purple dashed lines represent the calibrated time series from calibration scenario S4, which are taken from the runs that yield the best simulations.







**Figure 11.** NSE, KGE, RMSE and r variations comparing calibration scenarios S1, S2, S3 and S4 with S0. Rows show the three global precipitation products and columns show the performance metrics.





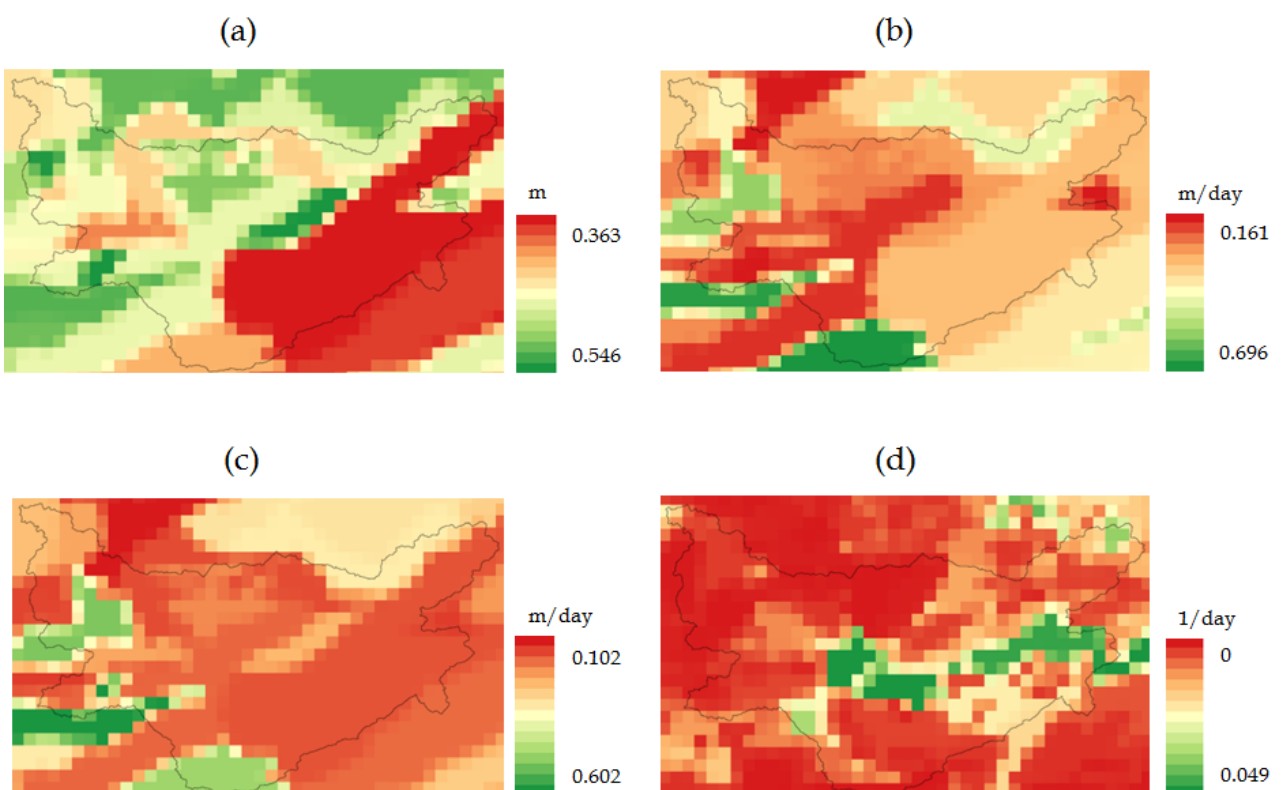

**Figure A1.** Initial model parameter values for the S0 calibration scenario (reference): (a) total soil water storage capacity ($W_{max} = SC_1 + SC_2 + SC_3$), (b) saturated hydraulic conductivity of the $1^{st}$ and $2^{nd}$ soil layers ($K_{sat1}$ and $K_{sat2}$), (c) saturated hydraulic conductivity of $3^{rd}$ soil layer ($K_{sat3}$) and (d) baseflow recession coefficient ($J$).





*Acknowledgements.* This research received funding from the European Union Seventh Framework Programme (FP7/2007-2013) under grant agreement no. 603608, Global Earth Observation for integrated water resource assessment: eartH2Observe. We would like to thank ICARDA for their engagement and collaboration during this work. For the discharge data, we are very grateful to Yves Tramblay from the Institut de Recherche pour le Developpement (IRD) in France. For the GLEAM evapotranspiration data, we would like to thank VU Amsterdam in The
5  Netherlands and Ghent University in Belgium. For the ESA CCI soil moisture, we are indebted to the Vienna University of Technology in Austria.