# Peer review of "Calibration of a large-scale hydrological model using satellite-based soil moisture and evapotranspiration products"

_Hydrology and Earth System Sciences, 2017_

## Referee Comment (RC1) · R. C. Nijzink (Referee) · 24 Jan 2017

The study of Lopez Lopez et al. deals with a modelling exercise carried out for the Oum Er Rbia basin. The hydrological model PCR-GLOBWB was applied for this basin and optimized with GLEAM evaporation and ESA CCI surface soil moisture, in different calibration scenarios. In addition, three different precipitation products were used as forcing data. The authors show that a step-wise calibration with GLEAM and ESA CCI and forcing data from EI and MSWEP provides improvements in model performances. The added value of this research is very clear to me. I fully agree with the authors that

applications of (remotely sensed) data in hydrological modelling are mostly limited to one extra model variable to calibrate on, whereas combinations of data products are explored to a much lesser extent. In addition, the paper is well written and concise. Nevertheless, I'd like to raise several remarks in order to help the authors to improve on their manuscript.

**General comments**

My most important point considers the calibration. It consists of 81 runs with three different values for the calibrated prefactors. I fully understand that running a distributed model has a high computation cost, but this number of runs seems rather low to me. There is a big risk of undersampling, leading to results that can just be a mere coincidence. For example, the most optimal value of $f_w$ may actually be 0.889, whereas only 0.75, 1 and 1.25 are explored in the study. Once again, I understand the burden of computational efforts, but at least the authors may want to reflect on this limitation in their discussion. In addition, it is mentioned that, except for these four prefactors, the remaining parameters were kept fixed. How many parameters are kept fixed and to what extend is the model already directed towards a certain solution by the choice of fixing certain parameters? For example, according to Figure 2 PCRGLOB-WB uses an interception routine. If the maximum interception capacity is kept fixed, it will probably influence the results for GLEAM versus the modelled evaporation.

In addition to this, the step-wise calibration consists out of first calibrating on evaporation with GLEAM, and, in a second step, on soil moisture. I just wonder how much influence this order in calibration influences the results, especially as soil moisture strongly influences evaporation. Did you consider a step-wise calibration with first ESA CCI soil moisture and then GLEAM in a second step?

I also wonder what the reasoning is behind the choice to compare the ESA CCI surface soil moisture with the soil moisture of the first three soil layers of PCR-GLOBWB. As mentioned by the authors (page 8, line 30) the ESA CCI soil moisture only represents

the first 0.5-2cm, so wouldn't it make more sense to just compare with only the very first soil layer (first 5cm according to page 5, line 14) in PCR-GLOBWB? In this way, all parameters affecting the soil moisture in all the three layers will react, which can also be noted from the results for $f_k$ , but one could wonder whether this is for the right reason.

Often, a comparison is made between the reference scenario S0 and the scenarios S2-S4. Nevertheless, S0 is merely an uncalibrated model and especially for Ait Ouchene (Figure 6), the model performances are rather poor. Therefore, not much is needed to achieve improvements in this case. Isn't it much more interesting to focus more on comparing S1 with S2-S4? In other words, how close can we get to a calibration on streamflow with help of GLEAM and ESA CCI? It would be interesting to see if differences occur in Figures 8-10 for S1 and S2-4. Ideally, there would be no difference, but I expect that this will not be the case.

**Detailed comments**

P8.L2-3. I don't know if these specific stations were used for MSWEP, but as MSWEP used station data as input (also remarked by the authors on P7.L19-20), isn't it logical that MSWEP provided a better fit to the station data?

P10.L15-17. Why model at a daily basis and only compare on monthly values? What is the temporal resolution of the data (discharge, GLEAM and ESA CCI)?

P12.L10. I can see that $f_w$ shows a clear pattern, but I don't see this clearly for $f_e$.

P13.L16-17. These numbers refer to the WFDEI-case

P14.L9-11. It may as well be model structural deficiencies as wrong parameterizations. It is a bit easy to blame the input data directly, especially as it happens for two out of three input products. It must noted as well that even though EI has the peaks in 2002 right, it also underestimates the peaks in 1996 and 1997.

Throughout the manuscript, the terms KGE, NSE etc. are used and sometimes refer to

a case with evaporation and sometimes to cases with soil moisture or discharge. For clarity, it might be good to add a subscript (e.g. $KGE_E$, $KGE_{SM}$ etc.).

---

## Referee Comment (RC2) · H. Bogena (Referee) · 25 Feb 2017

This paper deals with a classical calibration study of the large-scale hydrological model PCR-GLOBWB using satellite-based products of evapotranspiration and soil moisture by taking the Moroccan Oum Er Rbia basin as an example. This topic fits very well to the scope of this journal. However, at times the paper is difficult to comprehend and dull to read. Especially the presentation of the calibration results gave me a hard time. Also, the structuring of the results section needs to be improved (e.g. introduction of subchapters). The revised version should also be checked by a native speaker.

[Figure]

Notwithstanding its formal deficiencies, this paper gives some valuable information on the calibration of distributed models in data poor regions and the usage of satellite-based data products. Therefore, I believe the paper deserves to be published after addressing the issues with regards to contents and presentation as listed in detail in the following.

General comments:

The motivation for choosing the study area is too weak. Basically the research presented in this study could be accomplished in any catchment. For instance, you could mention the specific challenges for the calibration of hydrological models in such environments.

The introduction is repetitive and too long. Please rewrite in a more focussed way and describe more clearly the structure of the paper.

A justification for using 6 performance metrics for the precipitation evaluation is missing. Since only the performance metrics NSE and KGE are used for the model validation analysis, I suggest to limit the precipitation data evaluation also to these metrics.

The presentation of the results needs to be improved. It is very difficult to keep the attention to the text, because the text is difficult to comprehend and the results are merely listed. Also a critical in-depth discussion of the results is largely missing.

Specific comments:

P3L18: GLEAM is a comprehensive model for the estimation of terrestrial evaporation and root-zone soil moisture from satellite data. Please clarify.

P7L3-20: This section is copious and repetitive. Please rewrite in a more clear and concise way.

P7L32-P8L6: This section should be placed in the results section. Please indicate the temporal resolution of the data from the rainfall gauging stations. In addition it is not
clear to me, why you need to six performance metrics for the precipitation validation. Also, just listing the values of the metrics is not sufficient.

P9L3: Why did you use the first three layers? Given the extremely low penetration depth of the C-band data used for the ESA CCI SM product, you should only compare to the first layer. The depth of this layer needs to match the penetration depth of the C-band data, i.e. 2 cm.

P9L5-8: In my opinion this procedure leads to an untrustworthy and unsound comparison of simulated and observed soil moisture. A direct comparison of model results and observed data is a prerequisite for an unbiased and unadorned evaluation of the simulation results.

P9L15-34: This section is copious and repetitive. Please rewrite in a more clear and concise way.

P10L2: Why did you choose KGE for this analysis?

P10L4: In my view, it does not make sense to recalibrate for each precipitation data set. I would be more sensible to use the data that corresponds best with the rainfall gauging station data.

P10L5: I guess the different Ksat-values correspond to the soil layers S1-3. What about the Ksat-value for S4?

P10L6: Reference potential ET is not a parameter. Since it varies in time, it is variable.

P12L8: Delete "nearly"

P12L8: Due its complexity, Figure 5 is difficult to comprehend. In particular, the meaning of the dots in the scatterplots stays unclear with regard to what are actually representing (i.e. why is there more than one dot per variant). Since only three different values of the prefactors are considered no continuous scale should be used for the x-axis (otherwise the reader gets puzzled why the dots are not spreading).

P12L10: Please explain in more detail, why these prefactors should be well defined.

P12L16: "are considered"

P12L17: Please explain in more detail, why these prefactors should be well defined.

P12L23-31: This section is incomprehensible. Please rewrite.

P12L34: "scatterplots"

P13L3: The scatterplots of Figures 6 and 7 are quite repetitive. I would be enough to present only the NSE and KGE values of all variants and some selected scatterplots in case it is helpful for the discussion of the results.

P13L6-7: This is very obvious and provokes the question why you are using the poorer precipitation data sets for the model calibration analysis at all.

P13L26: You should first introduce the motivation for presenting these figures.

P13L35: I would rather like to see the unscaled results, because this procedure embellishes the model results.

P14L9-10: You could simply check using the measured precipitation data.

P14L30: See comment P13L6-7

P15L8-9: So wouldn't it be more sensible to use multi-objective function calibration procedure?

P15L10: The discussion chapter is largely a summary of the results (first half) and an outlook, which should be placed in the conclusion chapter. In order to reduce redundancy, I suggest skipping this chapter and moving parts to the results and conclusion chapters.

P16L12: Another promising way towards a more in-depth validation of distributed models are the empirical orthogonal functions analysis and the wavelet coherence analysis (e.g. Fang et al., 2015).

Figures:

Figure 3: The precipitation field seems to be shifted (the highest precipitation amounts are expected in the Atlas mountain ranges, see e.g. Chehbouni et al., 2008). You should add dots in the lower graphic.

Figure 5: The scatterplots are too crowed and difficult to read.

Figures 6-10: Always indicate that you are showing monthly values, e.g. "Monthly observed discharge".

Additional literature

Chehbouni, A., Escadafal, R., Boulet, G., Duchemin, B., Simonneaux, V., Dedieu, G., Mougenot, B., Khabba, S., Kharrou, M.H., Merlin, O., Chaponnière, A., Ezzahar, J., Erraki, S., Hoedjes, J., Hadria, R., Abourida, H., Cheggour, A., Raibi, F., Boudhar, A., Hanich, L.,Guemouria, N., chehbouni, Ah., Olioso, A., Jacob, F. and Sobrino, J. (2008): An integrated modelling and remote sensing approach for hydrological study in semi-arid regions: the SUDMED Program. International Journal of Remote Sensing, 29: 5161-5181.

Fang, Z., H.R. Bogena, S. Kollet, J. Koch and H. Vereecken (2015): Spatio-temporal validation of long-term 3D hydrological simulations of a forested catchment using empirical orthogonal functions and wavelet coherence analysis. J. Hydrol. 529: 1754-1767, doi:10.1016/j.jhydrol.2015.08.011.

---

## Short Comment (SC1) · 3 Mar 2017

This study caught my interest because it aims at improving discharge predictions in data scare areas using remote sensing (RS) data of evapotranspiration (ET) and soil moisture (SM) as calibration targets to inform model parameters of a distributed model. I think the work is nicely presented and a very relevant contribution to HESS. However, I would like to discuss one aspect, which the authors did not touch upon. This may inspire the authors when revising the manuscript.

I would like to ask the authors if they regard spatial averaging of RS data as the optimal

way to utilize these data? If I understand correctly from Figures 8 & 9, the authors used timeseries of averaged ET and SM as calibration targets. This neglects the valuable spatial information which is contained in the RS data. One could imaging that an additional metric, which is targeted at the spatial patterns of ET and SM may improve the discharge performance from calibration scenarios 2-4. A pattern oriented calibration can potentially yield more realistic parameter fields which are necessary to be able to simulate the spatial variability of runoff generation within the catchment in a more realistic manner.

Ultimately, if the spatial variability of hydrological processes is not of concern and the model is simply calibrated, evaluated and used at aggregated scale, then why not just use a simple lumped model? I think the discussion of the manuscript at hand would be significantly improved by showing and discussing simulated and observed spatial patterns of ET and SM.

---

## Author Comment (AC1) · 21 Apr 2017

**Response to short comment of Dr. Julian Koch on the manuscript "Calibration of a large-scale hydrological model using satellite-based soil moisture and evapotranspiration products"**

**Patricia López López, Edwin H. Sutanudjaja, Jaap Schellekens, Geert Sterk and Marc F. P. Bierkens**

The authors would like to thank Dr. Julian Koch for his time and useful comment on the manuscript. His suggestion will help us to improve the quality of our manuscript. We have included detailed responses to his comment in the supplementary .pdf file. We have also included a modified version of the original manuscript. Please note the supplement to this comment and the modified manuscript, with modifications in blue.

**Short comment:**

*This study caught my interest because it aims at improving discharge predictions in data scare areas using remote sensing (RS) data of evapotranspiration (ET) and soil moisture (SM) as calibration targets to inform model parameters of a distributed model.*
*I think the work is nicely presented and a very relevant contribution to HESS. However,*
*I would like to discuss one aspect, which the authors did not touch upon. This may inspire the authors when revising the manuscript.*
*I would like to ask the authors if they regard spatial averaging of RS data as the optimal way to utilize these data? If I understand correctly from Figures 8 & 9, the authors used*
*timeseries of averaged ET and SM as calibration targets. This neglects the valuable spatial information which is contained in the RS data. One could imaging that an additional metric, which is targeted at the spatial patterns of ET and SM may improve the discharge performance from calibration scenarios 2-4. A pattern oriented calibration can potentially yield more realistic parameter fields which are necessary to be able to simulate the spatial variability of runoff generation within the catchment in a more realistic manner.*
*Ultimately, if the spatial variability of hydrological processes is not of concern and the model is simply calibrated, evaluated and used at aggregated scale, then why not just use a simple lumped model? I think the discussion of the manuscript at hand would be significantly improved by showing and discussing simulated and observed spatial patterns of ET and SM.*

**Answer:**

As Dr. Koch pointed out in his comment, we used average actual evapotranspiration and surface soil moisture over the entire Oum Er Rbia basin to calibrate model parameters under five different scenarios. With this approach, we take into account the spatial variability of the catchment indirectly through the model parameters that are different for each grid cell. However, we agree with Dr. Koch that other calibration approaches could be applied to better use the spatial information from GLEAM and ESA CCI products. Therefore, we carried out a new calibration scenario based on ESA CCI surface soil moisture, but instead of comparing the basin average (calibration scenario S3), we compared soil moisture estimates and observations per grid cell (calibration scenario S3 pixel). This calibration approach allows the identifiability of the optimal prefactors set for each grid cell. This means that there is not a single spatially uniform prefactor (e.g. $f_w = 1.25$), but a map of different prefactor values

depending on the location / grid cell (e.g. $f_w$ (lat$_1$, lon$_1$) = 1.25, $f_w$ (lat$_2$, lon$_2$) = 1, etc.). A similar calibration strategy was described in Sutanudja et al. (2014).

Below, we included two figures (Figure RC1 and RC2) to summarize the results obtained with this new calibration scenario based on the pixel comparison of soil moisture model estimates and ESA CCI observations.

In Figure RC1 we plotted the prefactor values identified over the Oum Er Rbia basin in different columns (1[st] column: $f_e$, 2[nd] column: $f_j$, 3[rd] colum: $f_k$ and 4[th] column: $f_w$) for the three global precipitation products in different rows (top: EI, bottom: WFDEI and middle: MSWEP).

From Figure RC1, there is not a clear spatial pattern for each prefactor values. For example, $f_k$ values vary non-uniformly over the basin, with values of 0.25 in a location next to others with values of -0.25 and 0. This raises the question of how to regionalize the prefactor values and therefore, the model parameters. Further work would be needed to investigate novel strategies of regionalization based on land use, soil characteristics, climatic zones, etc.

In Figure RC2 we plotted discharge estimates and observations at Mechra Eddahk when MSWEP is used as model forcing. The red dashed lines represent discharge estimates from calibration scenario S0 (a) and S1 (b) and the purple dashed lines represent the calibrated time series from calibration scenario S3 pixel.

Similar to results obtained with the other calibration scenarios, discharge estimates reproduce ground observations quite well. Using ESA CCI soil moisture pixel by pixel for calibration (S3 pixel) improves the discharge estimates from 0.648 (S0, reference run) to 0.661 $KGE_q$ values (Figure RC2a). However, calibrating based on in-situ discharge observations (S1) leads to a further improvement to 0.680 (Figure RC2b). If we compare results of Figure RC2 with results of Figures 7 and 8 in the manuscript, $KGE_q$ values of 0.710 are reached when ESA CCI soil moisture and GLEAM actual evapotranspiration are used in the step-wise calibration approach.

With this new calibration scenario (S3 pixel), we are limiting the comparison to soil moisture estimates and observations pixel by pixel and problems of over-parameterization may occur, i.e. obtaining similar model results with more than one parameters combination. A possible route to overcome this problem could be using new spatial performance metrics to select the best parameters set (Koch et al., 2017).

We believe that the present experiment is useful and gives further insight into how to incorporate spatial information of variables such as ET and SM for calibration of spatially distributed hydrological models. However, we think that including these results in the manuscript may hamper the comprehension of the overall work. Therefore, we will discuss these aspects in section 5. Discussion and conclusions as follows:

P17L5: "… The spatial information of these satellite-based products could be used in a different way than the one explained in this study. For example, a calibration scenario based on a pixel by pixel, instead of basin average, comparison of surface soil moisture and actual evapotranspiration model estimates and observations could further improve discharge estimates. This calibration approach would have into account the spatial variability of the variables over the basin. Previous studies investigate how to incorporate spatial information into hydrological models using innovative spatial performance metrics to analyse the spatial sensitivity of simulated land-surface patterns (Koch et al., 2017). …"

**References:**

- Sutanudjaja, E. H., Van Beek, L. P. H., De Jong, S. M., Van Geer, F. C., & Bierkens, M. F. P. (2014). Calibrating a large-extent high-resolution coupled groundwater-land surface model using soil moisture and discharge data. *Water Resources Research*, *50*(1), 687-705.

- Koch, J., Mendiguren, G., Mariethoz, G., & Stisen, S. (2017). Spatial sensitivity analysis of simulated land-surface patterns in a catchment model using a set of innovative spatial performance metrics. Journal of Hydrometeorology, (2017).

**Figures:**

[Figure]

***Figure RC1.*** *Prefactors values identified with calibration scenario S3 pixel. Columns indicate different prefactors and rows indicate different global precipitation products.*

[Figure]

***Figure RC2.*** *Comparisons between monthly observed discharge (black) and estimated discharge (red and purple) time series at Mechra Eddahk when PCR-GLOBWB is forced with MSWEP precipitation. The red dashed lines represent discharge estimates from calibration scenario S0 (a) and S1 (b) and the purple dashed lines represent the calibrated time series from calibration scenario S3 pixel.*

---

## Author Comment (AC2) · 21 Apr 2017

**Response to review comments of Dr. Heye R. Bogena on the manuscript "Calibration of a large-scale hydrological model using satellite-based soil moisture and evapotranspiration products"**
**Patricia López López, Edwin H. Sutanudjaja, Jaap Schellekens, Geert Sterk and Marc F. P. Bierkens**

The authors would like to thank Dr. Heye R. Bogena for his really useful, valuable and productive suggestions on the manuscript. His in-depth review will help us to improve the structure and the overall quality of our manuscript. We have included detailed responses to his comments in the supplementary .pdf file. We have also included a modified version of the original manuscript. Please note the supplement to this comment and the modified manuscript, with modifications in blue.

**General comments:**

**Comment 1:**

*The motivation for choosing the study area is too weak. Basically the research presented in this study could be accomplished in any catchment. For instance, you could mention the specific challenges for the calibration of hydrological models in such environments.*

**Answer:**

We agree with the reviewer and according to his suggestion, we will include and modify some sentences about the specific challenges for hydrological modelling in ungauged basins as follows:

P2L4-9:"… Ungauged or poorly gauged river basins also include those basins where data are inaccurate, scarce, intermittent or collected at different temporal resolutions, leading to the problem that it is not clear how to integrate these data consistently into hydrological models (Winsemius et al., 2009). As a result, the limited availability and poor quality of data induces large uncertainty in model outputs from these river basins (Seibert and Beven, 2009). Developing novel strategies to enhance available datasets and hydrological models is one of the key strategies when working in ungauged basins (Hrachowitz et al., 2013). …"

We will also include more information about particularities of the Oum Er Rbia basin to improve the motivation behind the selection of this study area:

P3L8-13: "… with a good coverage of in-situ hydro-meteorological data. In the present study area, the Oum Er Rbia river basin located in Morocco, ground observations are spatially sparse and limited in number classifying it as a poorly-gauged river basin. The region frequently suffers from water scarcity and droughts and water availability is the main factor influencing socio-economic development, mostly driven by the agriculture (Houdret, 2009). The studies of Ouatiki et al. (2017), Tramblay et al. (2012) and Tramblay et al. (2016) are testimony to the relevance of this area. Therefore, developing new strategies to model this watershed is highly relevant to improve water management and assessment of the water availability within the basin. …"

**Comment 2:**

*The introduction is repetitive and too long. Please rewrite in a more focussed way and describe more clearly the structure of the paper.*

**Answer:**

We will modify the introduction to clarify the contents and structure of the manuscript as follows:

[revised manuscript text omitted]

**Comment 3:**

*A justification for using 6 performance metrics for the precipitation evaluation is missing. Since only the performance metrics NSE and KGE are used for the model validation analysis, I suggest to limit the precipitation data evaluation also to these metrics.*

**Answer:**

We agree with the reviewer that not all the performance metrics considered for precipitation evaluation are significant for the inter-comparison of the precipitation products. Therefore, and according to the reviewer's suggestion, we will reduce the number of performance metrics. We will use KGE as the main performance indicator. As KGE can be considered a weighted evaluation of NSE, r and Percent Bias (Gupta et al., 2009), we will also include the latter three metrics to analyse the importance of each component. These metrics will be added because similar values of KGE and NSE were found for EI and MSWEP precipitation datasets. Hence, r and Percent Bias will be used to analyse their differences in further detail. Better performance in terms of r was obtained with MSWEP and lower PBias values were found with EI. The analysis of these results, shown in Figure 4 (which will be also modified), is missing in the manuscript. We will modify the manuscript as follows (see comment 7):

P10L25-P11L3: "… Moreover, various performance metrics between the interpolated and in-situ ground data were calculated and shown in Figure 4. Overall, EI and MSWEP provide a better fit to the station data compared to WFDEI, with higher $KGE_{precip}$, $NSE_{precip}$ and $r_{precip}$ than WFDEI. When comparing EI with MSWEP, similar values of $KGE_{precip}$ and $NSE_{precip}$ were found, whereas higher differences exist in $r_{precip}$ and $PBias_{precip}$. In terms of correlation MSWEP shows the best performance, but EI shows the lowest Percent Bias at both weather stations, with a value of less than 10 % . Only two weather stations were found within the basin for the previous analysis. These measurements were considered too scarce to cover the basin and to discard the precipitation product with the lowest performance (WFDEI). Therefore, the three global precipitation products were used to calibrate PCR-GLOBWB under the five calibration scenarios. ..."

**Comment 4:**

*The presentation of the results needs to be improved. It is very difficult to keep the attention to the text, because the text is difficult to comprehend and the results are merely listed. Also a critical in-depth discussion of the results is largely missing.*

**Answer:**

We will improve section 4. Results in different ways to facilitate its reading and comprehension. Initially, we will structure it in different subsections, starting with the inter-comparison results of precipitation products (subsection 4.1.) (before placed in subsection 3.2. Data), following with calibration results (subsection 4.2.) and ending with validation results (subsection 4.3.). The structure will be as follows:

4. Results

    4.1. Inter-comparison of precipitation products (see comment 3 and 7)

    4.2. Calibration results

        4.2.1. Calibration using in-situ observed discharge time series (S1)

        4.2.2. Calibration using GLEAM actual evapotranspiration time series (S2)

        4.2.3. Calibration using ESA CCI surface soil moisture time series (S3)

        4.2.4. Step-wise calibration using GLEAM actual evapotranspiration and ESA CCI surface soil moisture time series (S4)

    4.3. Validation results

At the same time, section 5. Discussion will be deleted and incorporated into a last section called Discussion and conclusions, where the results will be critically discussed (see comment 29).

Moreover, figures will be improved (see comments 16, 31, 32 and 33) together with their analysis (see comments 17-28).

**Detailed comments (P: page, L: line or lines):**

**Comment 5:**

*P3L18: GLEAM is a comprehensive model for the estimation of terrestrial evaporation and root-zone soil moisture from satellite data. Please clarify*

**Answer:**

The review is right. Indeed the GLEAM is a model that estimates the different components of terrestrial evaporation. We will clarify this aspect in section 1. Introduction as follows:

P3L16-17: "… to streamflow data. We use the evapotranspiration product generated by an enhanced version of the GLEAM model (GLEAM v3.0; Martens et al., 2016b) in combination …"

This will be also clarified in section 3.2.3. Evapotranspiration data and in section 3.2.4. Soil moisture data as follows:

P7L6-10: "… 1980-2014. To generate the GLEAM evapotranspiration product, the GLEAM model separately estimates the different components of terrestrial evaporation, including transpiration, interception loss, bare-soil evaporation, snow sublimation and open-water evaporation. To this end, it consists of four modules: the evaporation module, the stress module, the soil-water balance module and the rainfall interception model (Martens et al., 2016a). GLEAM evapotranspiration ($0.25^{o}$ x $0.25^{o}$) was interpolated …"

P7L20: "…. Similarly to GLEAM evapotranspiration, ESA CCI SM product …"

**Comment 6:**

*P7L3-20: This section is copious and repetitive. Please rewrite in a more clear and concise way.*

**Answer:**

We agree with the reviewer's comment and this section will be rewrite as follows:

P6L9-27: "… The meteorological data required to force PCR-GLOBWB are air temperature, precipitation and reference potential evapotranspiration. Air temperature and precipitation were obtained from the WATCH Forcing Data methodology applied to ERA-Interim reanalysis data (WFDEI) at an original spatial resolution of $0.5^{o}$ x $0.5^{o}$ (Weedon et al., 2014). Reference potential evapotranspiration was obtained through the FAO Penman-Monteith equation. Precipitation, air temperature and reference potential evapotranspiration were downscaled from the original spatial resolution to a $0.08^{o}$ x $0.08^{o}$ grid. Precipitation and air temperature were downscaled using precipitation and temperature lapse rates derived from the 10' CRU-CL2.0 data (New et al., 2002) through a linear regression analysis (Sutanudjaja et al., 2011). Reference potential evapotranspiration was downscaled using the e2o-downscaling-tools (Schellekens and Sperna Weiland, 2017; Sperna Weiland et. al. 2015).

To test model sensitivity to precipitation, air temperature and reference potential evapotranspiration were fixed and two additional global precipitation products were used: (i) ERA-Interim reanalysis data (EI) from the European Centre for Mediumrange Weather Forecasts (ECMWF) at the original spatial resolution of $0.5^{o}$ x $0.5^{o}$ (Dee et al., 2011) and (ii) Multi-Source Weighted-Ensemble Precipitation data (MSWEP) by merging gauge, satellite and reanalysis data at the original spatial resolution of $0.25^{o}$ x $0.25^{o}$ (Beck et al., 2016c).

The three global precipitation products were inter-compared and interpolated to the two weather station locations found inside the Oum Er Rbia basin (http://www.wmo.int/pages/themes/climate/), Beni Mellal and Kasba Tadla (Figure 1). Nash-Sutcliffe efficiency (NSE), Kling-Gupta efficiency (KGE), Pearson's correlation coefficient (r) and Percent Bias (PBias) between the interpolated and in-situ ground daily data were calculated. A description of the performance metrics with their mathematical formulation is

included in section 3.4. These metrics were selected to have detailed information about differences between precipitation products. …"

**Comment 7:**

*P7L32-P8L6: This section should be placed in the results section. Please indicate the temporal resolution of the data from the rainfall gauging stations. In addition it is not clear to me, why you need to six performance metrics for the precipitation validation. Also, just listing the values of the metrics is not sufficient*

**Answer:**

We think that moving this subsection to section 4. Results is a good suggestion to improve the structure and analysis of the results. Following the reviewer's suggestion the inter-comparison of the three global precipitation products will be placed in section 4. Results, subsection 4.1. Inter-comparison of precipitation products (see comment 4). We will indicate the daily temporal resolution of the precipitation validation (P6L29). We will reduce the number of performance metrics and we will improve the analysis of these metrics (see comment 3). Therefore, this section will be modified as follows:

P10L16-P11L2: "4.1. Inter-comparison of precipitation products

To inter-compare the precipitation products, the annual mean precipitation for the study time period (1979-2010) for each forcing dataset was calculated (Figs. 3a, 3b and 3c). In addition to the spatial resolution difference, MSWEP is able to capture the rainfall pattern over the Atlas Mountains rather well, which is only roughly distinguished by WFDEI and unrecognized by EI. The finer spatial resolution and the combination of reanalysis, satellite and in-situ data are probably the reasons for its more plausible spatial pattern. Furthermore, climatology of precipitation products was analyzed (Fig 3d). WFDEI ranges from 4.5 mm in July to 57 mm in February, whereas EI and MSWEP show a lesser variability with precipitation values from 10.5 mm in July to 42.6 mm in November. Smaller differences between WFDEI and EI and MSWEP are observed during the summer months. EI and MSWEP show similar temporal precipitation patterns. Annual mean precipitation over the entire basin obtained with MSWEP (355.15 mm) is approximately 80 mm higher than with EI (276.67 mm). Similar annual median values are obtained with the three global precipitation products, although the distribution of WFDEI highly differs from the other two products.

Moreover, various performance metrics between the interpolated and in-situ ground data were calculated and shown in Figure 4. Overall, EI and MSWEP provide a better fit to the station data compared to WFDEI, with higher $KGE_{precip}$, $NSE_{precip}$ and $r_{precip}$ than WFDEI. When comparing EI with MSWEP, similar values of $KGE_{precip}$ and $NSE_{precip}$ were found, whereas higher differences exist in $r_{precip}$ and $PBias_{precip}$. In terms of correlation MSWEP shows the best performance, but EI shows the lowest Percent Bias at both weather stations, with a value of less than 10 % . Only two weather stations were found within the basin for the previous analysis. These measurements were considered too scarce to cover the basin and to discard the precipitation product with the lowest performance (WFDEI). Therefore, the three global precipitation products were used to calibrate PCR-GLOBWB under the five calibration scenarios."

**Comment 8:**

*P9L3: Why did you use the first three layers? Given the extremely low penetration depth of the C-band data used for the ESA CCI SM product, you should only compare to the first layer. The depth of this layer needs to match the penetration depth of the C-band data, i.e. 2 cm.*

**Answer:**

According to P7L27-28: "ESA CCI surface soil moisture observations were compared to simulated soil moisture with the first of the three vertical soil layers in PCR-GLOBWB." To clarify this and following the reviewer's suggestion, we will include a note as follows:

P7L23-24: "… ESA CCI surface soil moisture observations were compared to simulated soil moisture of the first of the three vertical soil layers in PCR-GLOBWB (top 5 cm of soil). …"

**Comment 9:**

*P9L5-8: In my opinion this procedure leads to an untrustworthy and unsound comparison of simulated and observed soil moisture. A direct comparison of model results and observed data is a prerequisite for an unbiased and unadorned evaluation of the simulation results.*

**Answer:**

Following the reviewer's suggestion, we produced two figures with the original and the rescaled simulated soil moisture time series before and after the mean-standard deviation matching technique is applied (see comment 25). This rescaling approach applied to surface soil moisture have been previously used in several studies to overcome the existent uncertainties in satellite observations and model estimates (Koster et al., 2009; Renzullo et al., 2014; Su et al., 2013). However, we agree with the reviewer that other possible approaches could have been investigated including an analysis of the optimal soil depth in the model corresponding to the depth for satellite measurements. Due to computational time limitations and to avoid numerical stability problems derived from the daily temporal resolution, we decided to follow a mean-standard deviation matching technique.

**Comment 10:**

*P9L15-34: This section is copious and repetitive. Please rewrite in a more clear and concise way.*

**Answer:**

We will modify the section as follows:

P8L5-19: "… Alternative single objective calibration approaches based on discharge, actual evapotranspiration and surface soil moisture and a multiobjective calibration approach based on actual evapotranspiration and surface soil moisture were inter-compared. Five different calibration scenarios were carried out. Calibration scenario S0 represents the reference calibration scenario, which was not locally calibrated for the Oum Er Rbia basin, but uses a-priori model parameters derived from vegetation, soil properties and geological information at a global scale (latest model version of PCR-GLOBWB). Calibration scenario S1 aims to calibrate the hydrological model using in-situ discharge observations, following the traditional calibration approach. Calibration scenarios S2 and S3 use GLEAM actual

evapotranspiration and ESA CCI surface soil moisture time series for calibration, respectively. Calibration scenario S4 represents the multiobjective calibration approach and it consists of a step-wise calibration scheme that attempts to combine the strengths of calibration scenarios S2 and S3. Step one is simply scenario S2, where all the model parameters are allowed to be adjusted based on GLEAM actual evapotranspiration. In step two, those parameters that are clearly identified by calibration scenario S2 are held constant and the remaining parameters are allowed to be adjusted according to ESA CCI surface soil moisture, calibration scenario S3.

The five calibration scenarios were analysed for each of the three global precipitation products to study their impact on model parameters calibration and model performance. The calibration scenarios are described in Table 2, including the scenario identifier. …"

**Comment 11:**

*P10L2: Why did you choose KGE for this analysis?*

**Answer:**

Traditional calibration and evaluation approaches of hydrological models with observed data use Mean Squared Error (MSE) and Nash-Sutcliffe efficiency (NSE) as the objective functions to maximize. However, Gupta et al., (2009) proposed Kling-Gupta efficiency (KGE) as an alternative criterion to avoid possible problems derived from the use of NSE, such as the underestimation of high values and overestimation of low values. Moreover, KGE can be considered a weighted evaluation of NSE, r and Percent Bias. Based on this, we decided to choose KGE for the analysis of calibration results. We will add a sentence explaining the selection of KGE as follows:

P8L21-23: "… for the calibration scenarios was Kling-Gupta efficiency (KGE), instead of the traditional Mean Squared Error (MSE) or Nash Sutcliffe efficiency (NSE) to avoid underestimating the variability of values (Gupta et al., 2009). The mathematical …"

We will also modify the manuscript as follows:

P15L17-24: "… A possible route to overcome this problem may be to use various performance indicators (for example, KGE, NSE, PBias and r) as objective functions to optimize in each calibration scenario, instead of using a single one. This multiobjective calibration approach may further improve discharge model estimates. …"

**Comment 12:**

*P10L4: In my view, it does not make sense to recalibrate for each precipitation data set. I would be more sensible to use the data that corresponds best with the rainfall gauging station data.*

**Answer:**

As it is indicated in P8L17-19, the analysis of the five calibration scenarios for each global precipitation product, allow us to study their impact on model parameters calibration and model performance. Moreover, we can analyse the influence and/or importance on model performance of the calibration approach in comparison with the precipitation dataset used as forcing.

Furthermore, from the inter-comparison of precipitation products included in section 4.1 (see comments 3 and 7) it was not possible to select only one precipitation product, because MSWEP performed better for some indicators and worse for others in comparison with EI.

On the other hand, only two rainfall stations were found inside the Oum Er Rbia basin, as it is mentioned in section 3.2.1. Meteorological data. These measurements were considered too scarce in number and spatially sparse to cover the entire basin and therefore, to select the best global precipitation product and discard the remaining ones.

We will add a sentence on this aspect in section 4.1. Inter-comparison of precipitation products as follows:

P10L29P11L5-2: "… than 10 % . Only two weather stations were found within the basin for the previous analysis. These measurements were considered too scarce to cover the basin and to discard the precipitation product with the lowest performance (WFDEI). Therefore, the three global precipitation products were used to calibrate PCR-GLOBWB under the five calibration scenarios. ..."

**Comment 13:**

*P10L5: I guess the different Ksat-values correspond to the soil layers S1-3. What about the Ksat-value for S4?*

**Answer:**

Indeed, we also calibrated the $K_{sat4}$, with the calibration of the baseflow recession coefficient (J). Soil in PCR-GLOBWB is divided into three vertical layers representing the top 5 cm of soil (S1), the following 25 cm of soil (S2) and the remaining 120 cm of soil (S3). Under the third soil layer, there is a groundwater store (S4). The saturated hydraulic conductivities of 1$^{st}$, 2$^{nd}$ and 3$^{rd}$ soil layers are $K_{sat1}$, $K_{sat2}$ and $K_{sat3}$, which controls the vertical fluxes between soil layers and the groundwater store, affecting the groundwater recharge. Baseflow from the active groundwater layer depends on the baseflow recession coefficient (J), which varies in function of the aquifer transmissivity and the aquifer specific yield. Therefore, J is included as a model parameter to calibrate. J is a recession coefficient parameterized based on Kraaijenhoff van de Leur (1958):

$$J = \frac{\pi^2 (KD)}{4 S_y L^2}$$

With $KD$ and $S_y$ indicating aquifer transmissivities and specific yields, and L indicating average flow lengths.

**Comment 14:**

*P10L6: Reference potential ET is not a parameter. Since it varies in time, it is variable.*

**Answer:**

Indeed, reference potential evapotranspiration is a variable and not a model parameter. We calibrated three model parameters and in addition, we also checked the uncertainty of

reference potential evapotranspiration following a similar approach with prefactors. We will modify the manuscript to clarify this topic as follows:

P8L25-P9L3: "… To calibrate PCR-GLOBWB for each of the three precipitation products, 81 runs with different parameter values were simulated: minimum soil water capacity ($W_{min}$), soil saturated hydraulic conductivites ($Ksat_1$, $Ksat_2$ and $Ksat_3$) and baseflow recession coefficient (J). These model parameters, which vary spatially over the basin, influence different model parts of the model behaviour, as it was explained in section 3.1. For the variation of the parameter values, spatially uniform prefactors were used: $f_w$, $f_k$ and fj (Table 3). The remaining model parameters were kept fixed.

The prefactors to vary model parameter values were referred to the parameters of the S0 calibration scenario. The spatial distribution of the parameters $W_{min}$, $K_{sat}$ and J used in S0 scenario can be found in Figure A1 of Appendix A.

Furthermore, the uncertainty of reference potential evapotranspiration ($E_{p,0ref}$) was also investigated using a correction prefactor, $f_e$, to this model variable. Considered values for $f_e$ prefactor are included with the previously mentioned ones in Table 3.

As reference calibration scenario, S0 prefactors are: $f_w=1$, $f_k=0$, $f_j=1$ and $f_e=1$. The model performances of all the simulations were evaluated for each of the five calibration scenarios to identify the best prefactor sets as the calibrated prefactor sets. …"

**Comment 15:**

*P12L8: Delete "nearly"*

**Answer:**

We will delete "nearly".

**Comment 16:**

*P12L8: Due its complexity, Figure 5 is difficult to comprehend. In particular, the meaning of the dots in the scatterplots stays unclear with regard to what are actually representing (i.e. why is there more than one dot per variant). Since only three different values of the prefactors are considered no continuous scale should be used for the x-axis (otherwise the reader gets puzzled why the dots are not spreading).*

**Answer:**

We agree with the reviewer that Figure 5 is quite complex and difficult to explain and therefore, to understand. We will improve this figure in different ways: we will use different colours and dot shapes to indicate different values of $f_e$, we will modify the horizontal axis of each scatterplot limiting the tick marks and numbers to the values of the used calibration prefactors and we will change the label of y-axis to indicate when KGE values are based on discharge, actual evapotranspiration and surface soil moisture using subscripts $KGE_q$, $KGE_{evap}$ and $KGE_{sm}$. Moreover, to facilitate the comprehension of the scatterplot, we will modify the figure explanation as follows:

P11L5-15: "… Model parameters were calibrated using discharge, evapotranspiration and soil moisture observations through five different calibration scenarios for the time period

1981-1993. Figure 5 shows results of all runs produced in this study for different calibration scenarios based on: in-situ discharge observations (S1) at Ait Ouchene (Figure 5a) and Mechra Eddahk (Figure 5b), GLEAM actual evapotranspiration (S2, Figure 5c) and ESACCI surface soil moisture (S3, Figure 5d). For each sub-figure in Figure 5, KGE results (y-axis) of using the three precipitation products are plotted in different rows (top: EI, middle: WFDEI and bottom: MSWEP) and prefactor values are plotted in different columns (x-axis, 1$^{st}$ column: $f_e$, 2$^{nd}$ column: $f_j$, 3$^{rd}$ column: $f_k$ and 4$^{th}$ column: $f_w$). Each scatterplot contains 81 dots representing each run with a different combination of parameter values. This means that the KGE values are the same in the four scatterplots of a row (y- axis), but in each of these scatterplots, they are plotted against a different prefactor (x-axis). With Figure 5, prefactor, and therefore parameter, ranges leading to better and worse performances can be distinguished, as well as their global optimal values. If no optimal value can be inferred, prefactors from the calibration scenario S0 are maintained ($f_e$=1, $f_j$=0, $f_k$=0 and $f_w$=1) …"

**Comment 17:**

*P12L10: Please explain in more detail, why these prefactors should be well defined.*

**Answer:**

According to the reviewer's comment, we will improve the explanation of prefactors identifiabilities as follows:

P11L26-30: "… Figures 5a and 5b (calibration scenario S1) are similar. From these figures, $f_e$ (1$^{st}$ column) and $f_w$ (4$^{th}$ column) are well identified by discharge calibration at both gauging stations when forced with any of the three precipitation products. $f_e$ = 1.25 and $f_w$ = 1.25 lead to the highest $KGE_q$ values. However, it is not possible to identify the best prefactors of $f_j$ (2$^{nd}$ column) and $f_k$ (3$^{rd}$ column). There are no clear and distinct maximum values in the scatterplots of these figures, hence $f_j$ = 0 and $f_k$ = 0 are used. …"

**Comment 18:**

*P12L16: "are considered"*

**Answer:**

We will modify this sentence as follows:

P12L21-23: "… Therefore, model run with prefactors $f_e$ = 1.25, $f_j$ = 0, $f_k$ = 0 and $f_w$ = 1 is considered as the calibrated run based on the evapotranspiration performance. …"

**Comment 19:**

*P12L17: Please explain in more detail, why these prefactors should be well defined.*

**Answer:**

Similarly to comment 17, we will include a note as follows:

P12L19-21: "… Figure 5c (calibration scenario S2) indicates that only prefactor $f_e$ (1$^{st}$ column) can be clearly identified (the highest $KGE_{evap}$ values are obtained with $f_e$=1.25),

whereas the remainder of the prefactors ($f_j$, $f_w$ and $f_k$) are non identifiable, suggesting that evapotranspiration-based calibration may be unreliable in their identification. Therefore, …"

**Comment 20:**

*P12L23-31: This section is incomprehensible. Please rewrite.*

**Answer:**

We will modify this section as follows:

P13L11-19: "…Calibration scenario S4 attempts to combine the strengths of scenarios S2 and S3. In the first step, the model is calibrated using GLEAM evapotranspiration (S2, Figure 5c). From Figure 5c, only $f_e$ prefactor is well identified (the highest $KGE_{evap}$ value is obtained with $f_e = 1.25$). In the second step, $f_e$ prefactor that has been identified was held constant and the remaining three prefactors were allowed to be calibrated according to ESA CCI soil moisture (S3, Figure 5d). From Figure 5d, $f_w$ and $f_k$ are identifiable (the highest $KGE_{sm}$ values are obtained with $f_w = 1.25$ and $f_k = 0.25$). As a result, for calibration scenario S4, the prefactors identified during the evapotranspiration calibration (S2): $f_e = 1.25$ and during the soil moisture calibration (S3): $f_w = 1.25$ and $f_k = 0.25$ are adopted. This step-wise calibration approach using multiple system variables allow to identify more parameters than when those variables are separately used. Nonetheless, neither of the steps in calibration scenario S4 allow the clear identification of $f_j$, so its value for the calibration scenario S0 is used, $f_j = 0$. …"

**Comment 21:**

*P12L34: "scatterplots"*

**Answer:**

We will correct "scatteplots" to "scatterplots".

**Comment 22:**

*P13L3: The scatterplots of Figures 6 and 7 are quite repetitive. I would be enough to present only the NSE and KGE values of all variants and some selected scatterplots in case it is helpful for the discussion of the results.*

**Answer:**

We agree with the reviewer on this matter. We believe that scatterplots of Figures 6 and 7 are helpful for the analysis and the discussion of the results. During the writing of the manuscript, the authors considered to delete Figure 6 and include only scatterplots for Mechra Eddahk station (Figure 7). However, this may give the impression that both discharge stations performs similarly for all calibration scenarios and precipitation products, which is not true. Therefore, we will move Figure 6 from the manuscript and include it to the Supplementary Information.

We will also improve the explanation of Figure 6 and Figure 2 in the Supplementary Information and we will modify the analysis of calibration results of these figures as follows:

P11L16-21: "… Once the best runs for each calibration scenario were identified, their discharge performance was checked at the two gauging stations: Mechra Eddahk, in Figure 6, and Ait Ouchene, in Figure 2 of the Supplementary Information. Observed discharge (y-axis) and estimated discharge (x-axis) are plotted in Figure 6 for the five calibration scenarios. Different rows in Figure 6 indicate the three global precipitation products (top: EI, middle: WFDEI and bottom: MSWEP) and different columns indicate the five calibration scenarios (1$^{st}$ column: S0, 2$^{nd}$ column: S1, 3$^{rd}$ column: S2, 4$^{th}$ column: S3 and 5$^{th}$ column: S4). The performance indicators NSE and KGE for discharge are included in every scatterplot in Figure 6 (NSE$_q$ and KGE$_q$). …"

P11L31-8: "… From Figure 6 (2nd column), the highest discharge performance is obtained when the model is calibrated with in-situ discharge observations (S1).

For all the calibration scenarios, a few general observations can be made. Scatterplots (Figure 6) highlight an overall better agreement and a lower bias between discharge observations and estimates for the Ait Ouchene (see Figure 2 in the Supplementary Information) than for Mechra Eddahk station. KGEq values at Ait Ouchene station for calibration scenario S0 are lower than for Mechra Eddahk station. …"

P12L24-26: "… From Figure 6 (3$^{rd}$ column), results indicate an increase of KGE$_q$ and NSE$_q$ values when GLEAM evapotranspiration is used for model calibration compared to the reference scenario (S0, 1$^{st}$ column of Figure 6). However, higher model performance values are obtained when calibrating based on in-situ discharge observations (S1, 2$^{nd}$ column of Figure 6). …"

P13L1-4: "… From Figure 6 (4$^{th}$ column), scatterplots indicate an improvement in the correspondence between observed and estimated discharge compared to the non-calibrated scenario (S0, 1$^{st}$ column of Figure 6). Similarly to calibration scenario S2 (3$^{rd}$ column of Figure 6), this improvement is lower than when the model is calibrated based on ground discharge observations (S1, 2$^{nd}$ column of Figure 6). …"

**Comment 23:**

*P13L6-7: This is very obvious and provokes the question why you are using the poorer precipitation data sets for the model calibration analysis at all.*

**Answer:**

This aspect has been already addressed in comment 12. Only two rainfall stations were found inside the Oum Er Rbia basin. These measurements were considered too scarce in number and spatially sparse to cover the entire basin and therefore, to select the best global precipitation product and discard the remaining ones. We will modify the manuscript as follows:

P12L7-16: "… Scatterplots (Figure 6) also show that estimated discharges are closer to observed discharges at both gauging stations when PCR-GLOBWB is forced with EI precipitation. Moreover, scatterplots indicate a worse agreement and a tendency to overestimate discharge when WFDEI and MSWEP are used. KGE$_q$ values for the reference calibration scenario S0 at Mechra Eddahk are 0.607, 0.325 and 0.561 when EI, WFDEI and MSWEP are used as forcing data respectively. These performance discrepancies are related with the differences between EI, WFDEI and MSWEP precipitation products discussed in

section 4.1. The lower quality of WFDEI in this region compared with the other precipitation datasets may be a possible reason of the lower discharge performance. When MSWEP was compared with in-situ precipitation data, performance in terms of correlation was higher than EI. However, EI showed less bias. The higher performance of discharge estimates when PCR-GLOBWB is forced with EI may be due to this bias difference and that the validation is carried out at a monthly temporal resolution, reducing the impact of correlation. …"

**Comment 24:**

*P13L26: You should first introduce the motivation for presenting these figures.*

**Answer:**

According to the reviewer's comment and considering the new structure of the results section, we will include a short paragraph explaining the motivation of Figures 7 and 8. We will also modify the analysis of Figures 7 and 8 as follows:

P13L28-P14L9: "… Once the model had been calibrated for each calibration scenario and each precipitation product, comparisons between estimates (before and after the calibration) and observations of actual evapotranspiration, surface soil moisture and discharge were carried out for the validation time period (1994-2011). To perform the analysis of these results, time series plots are included in Figures 7 and 8.

In Figure 7a, simulated actual evapotranspiration time series of the reference run (S0, red dashed line) and the step-wise calibrated run (S4, purple dashed line) are plotted against GLEAM actual evapotranspiration observations (black line). Similarly as Figure 7a, Figure 7b shows simulated surface soil moisture of the reference run (S0, red dashed line) and the step-wise calibrated run (S4, purple dashed line) plotted against ESA CCI surface soil moisture time series (black line). The rescaled soil moisture time series (after mean-standard deviation matching technique applied, see section 3.2.4) are shown. In Figure 7c, estimated discharge of the reference run (S0, red dashed line) and the step-wise calibrated run (S4, purple dashed line) are plotted against discharge observations (black line) at Mechra Eddahk. KGE values for actual evapotranspiration, surface soil moisture and discharge are included in Figures 7a, 7b and 7c. For the sake of simplicity, only results when the model is forced with MSWEP precipitation are shown.

Similarly to Figure 7, Figure 8 shows simulated evapotranspiration (Figure 8a), surface soil moisture (Figure 8b) and discharge (Figure 8c) against observations. However, in Figure 8, estimates of the discharge-calibrated run (S1, red dashed line) and the step-wise calibrated run (S4, purple dashed line) are plotted against observations (black line). …"

**Comment 25:**

*P13L35: I would rather like to see the unscaled results, because this procedure embellishes the model results.*

**Answer:**

According to the reviewer's suggestion, we produced two figures with the original and the rescaled simulated soil moisture time series before and after the mean-standard deviation matching technique is applied.

(a)

[Figure]

(b)

From these figures, the bias correction is observed between the rescaled and the original soil moisture time series. However, the inclusion of this fourth line (non-rescaled soil moisture) in the time series graphs of Figures 7 and 8 would difficult their interpretation and we believe that adding a new figure to the manuscript with original soil moisture time series would not improve the results analysis. Therefore, we will include it in Figure 1 of the Supplementary Information. We will also modify the manuscript as follows:

P8L1-3: "… When comparing the original and the rescaled soil moisture, it is observed that the mean-standard deviation technique effectively removes the biases between the simulated and observed soil moisture time series (see Figure 1 of the Supplementary Information). …"

**Comment 26:**

*P14L9-10: You could simply check using the measured precipitation data.*

**Answer:**

Only two rainfall stations were found inside the Oum Er Rbia basin. These measurements were considered too scarce in number and spatially sparse to cover the entire basin and therefore, to check the global precipitation products (see comments 12 and 23). Moreover, the lack of fit can be also due to model structural deficiencies. We will modify this section as follows:

P14L15-17: "… 1997. This lack of fit may be due to errors in the precipitation data, because higher discharge differences are shown when WFDEI and MSWEP products are used in comparison to EI product. Other possible reasons may be related with model structural deficiencies. When…"

**Comment 27:**

*P14L30: See comment P13L6-7*

**Answer:**

According to the reviewer's comment, we will delete the sentence: "This is a consequence of the precipitation discrepancies analysed in section 3.2.1."

**Comment 28:**

*P15L8-9: So wouldn't it be more sensible to use multi-objective function calibration procedure?*

**Answer:**

A multiobjective calibration approach using various objective functions, such as KGE, NSE, NSE for low flows, NSE for high flows, etc., may be an alternative route to calibrate model parameters. We will include a short paragraph on this topic in section 4.3. Validation results as follows:

P15L17-19: "… in terms of discharge. A possible route to overcome this problem may be to use various performance indicators (for example, KGE, NSE, RMSE and r) as objective functions to optimize in each calibration scenario, instead of using a single one. This multiobjective calibration approach may further improve discharge model estimates. …"

We will also discuss this topic in section 5. Discussion and conclusions as follows:

P16L5-6: "… a multiobjective calibration approach to streamflow observations could be followed. Similarly to Fenicia et al. (2007), multiple objective functions may be optimized in sequential steps for high flows, low flows and timing. …"

**Comment 29:**

*P15L10: The discussion chapter is largely a summary of the results (first half) and an outlook, which should be placed in the conclusion chapter. In order to reduce redundancy, I suggest skipping this chapter and moving parts to the results and conclusion chapters.*

**Answer:**

We believe that the reviewer is right and we will modify these sections. We consider that there are aspects, such as the possibility of other calibration approaches: multiobjective calibration, scaling relationships, catchments classification schemes, etc. or the potential use of other satellite products for hydrological modelling that are of interest of discussion. Therefore, and following the reviewer's suggestion, we will delete section 5. Discussion and we will modify section 6. Summary and conclusions to Discussion and conclusions to avoid repetitions as follows:

"5. Discussion and conclusions

This study investigates alternative routes to calibrate the large-scale hydrological model PCR-GLOBWB using earth observations globally available for the data-poor river basin of Oum Er Rbia in Morocco. Three global precipitation products, EI, WFDEI and MSWEP, are inter-compared and applied to force PCR-GLOBWB. Five different calibration scenarios are followed where GLEAM actual evapotranspiration and ESA CCI surface soil moisture data are used to identify model parameters with the aim to improve discharge estimates. In-situ discharge observations are also used for calibration, as they are traditionally used to calibrate hydrological models.

Results show that GLEAM actual evapotranspiration and ESA CCI soil moisture observations may be used to calibrate determined PCR-GLOBWB model parameters. GLEAM actual evapotranspiration can be used to calibrate the reference potential evapotranspiration ($f_e$), affecting the water exchange between the top soil layer and the atmosphere and hence the soil water balance. ESA CCI soil moisture data can be used to calibrate the minimum soil water capacity ($f_w$) and the saturated hydraulic conductivities of the soil layers ($f_k$), determining the surface runoff generation response, the shallow sub-surface flow and the groundwater recharge. However, calibration using only GLEAM evapotranspiration or only ESA CCI soil moisture can result in more than one parameters combination to be optimal in terms of discharge (overparametrization or equifinality problem). To overcome this problem, a step-wise calibration scenario based on both observations, evapotranspiration and soil moisture, can be included, allowing the identification of the optimal values of $f_e$, $f_w$ and $f_k$. Nonetheless, neither of these observations can be used to calibrate the baseflow from the active groundwater layer ($f_j$). To identify baseflow recession coefficient parameter ($f_j$) a multiobjective calibration approach to streamflow observations could be followed. Similarly to Fenicia et al. (2007), multiple objective functions may be optimized in sequential steps for high flows, low flows and timing.

[revised manuscript text omitted]

**Comment 30:**

*P16L12: Another promising way towards a more in-depth validation of distributed models are the empirical orthogonal functions analysis and the wavelet coherence analysis (e.g. Fang et al., 2015).*

**Answer:**

We will include a paragraph about alternative ways for validation of distributed hydrological models (see comment 29):

P16L15-18: "… Furthermore, the validation of this study was carried out exclusively on streamflow. Other validation approaches, including the empirical orthogonal functions, wavelet analysis or their combination, may be another promising way towards a more in-depth validation of distributed hydrological models (Mascaro et al., 2015; Koch et al., 2015; Fang et al., 2015) …"

**Figures:**

**Comment 31:**

*Figure 3: The precipitation field seems to be shifted (the highest precipitation amounts are expected in the Atlas mountain ranges, see e.g. Chehbouni et al., 2008). You should add dots in the lower graphic.*

**Answer:**

We will correct the precipitation shift mistake in Figures 3a, 3b and 3c. We will include dots in Figure 3d and therefore, produce a new complete Figure 3.

**Comment 32:**

*Figure 5: The scatterplots are too crowed and difficult to read.*

**Answer:**

This comment has been already addressed in comment 16.

**Comment 33:**

*Figures 6-10: Always indicate that you are showing monthly values, e.g. "Monthly observed discharge".*

**Answer:**

According to the reviewer's comment, we will modify Figures 5, 6, 7, 8 and 9 to indicate that the temporal resolution is always monthly.

**Additional modifications**

For the results analysis consistency of the manuscript, we will replace RMSE with PBias in Figure 9. The text will be also modified accordingly.

**Additional modifications in figures and figures to be included**

[Figure]

**Figure 3.** *(a) EI annual mean precipitation, (b) WFDEI annual mean precipitation and (c) MSWEP annual mean precipitation for 1979-2010 time period and (d) climatology of EI, WFDEI and MSWEP precipitation products.*

[Figure]

*Figure 4. Performance metrics of daily EI, WFDEI and MSWEP precipitation products at Beni Mellal and Kasba Tadla weather stations, including Kling-Gupta efficiency (KGE), Nash-Sutcliffe efficiency (NSE), Pearson's correlation coefficient (r) and Percent Bias (PBias).*

[Figure]

***Figure 5.*** *Scatterplots of discharge performance indicator KGE based on the monthly observations versus prefactors $f_e$, $f_j$, $f_k$ and $f_w$ for the calibration scenarios S1 ((a) Ait Ouchene (b) Mechra Eddahk), S2 (c) and S3 (d). In each sub-figure, columns indicate the different calibrated prefactors and rows indicate the three global precipitation products used as model forcing. Different colours and dot shapes indicate different $f_w$ values.*

[Figure]

***Figure 7.*** *(a) Monthly GLEAM actual evapotranspiration (black) and estimated actual evapotranspiration (red and purple) time series. (b) Monthly ESA CCI soil moisture (black) and estimated soil moisture (red and purple) time series. (c) Monthly observed discharge (black) and estimated discharge (red and purple) time series. The red dashed lines represent estimates from calibration scenario S0 (reference scenario). The purple dashed lines represent the calibrated time series from calibration scenario S4 which are taken from the runs that yield the best simulations. Estimated time series over the entire Oum Er Rbia basin for the validation time period obtained with MSWEP precipitation are shown.*

[Figure]

***Figure 8.*** *(a) Monthly GLEAM actual evapotranspiration (black) and estimated actual evapotranspiration (red and purple) time series. (b) Monthly ESA CCI soil moisture (black) and estimated soil moisture (red and purple) time series. (c) Monthly observed discharge (black) and estimated discharge (red and purple) time series. The red dashed lines represent estimates from calibration scenario S1. The purple dashed lines represent the calibrated time series from calibration scenario S4 which are taken from the runs that yield the best simulations. Estimated time series over the entire Oum Er Rbia basin for the validation time period obtained with MSWEP precipitation are shown.*

[Figure]

*Figure 9. KGE, NSE, r and PBias variations comparing monthly discharge estimates of calibration scenarios S1, S2, S3 and S4 with S0. Rows indicate the three global precipitation products and columns indicate the performance metrics.*

---

## Author Comment (AC3) · 21 Apr 2017

**Response to review comments of Remko C. Nijzink on the manuscript "Calibration of a large-scale hydrological model using satellite-based soil moisture and evapotranspiration products"**
**Patricia López López, Edwin H. Sutanudjaja, Jaap Schellekens, Geert Sterk and Marc F. P. Bierkens**

The authors would like to thank Remko C. Nijzink for his time and constructive and valuable comments on the manuscript. His suggestions will help us to improve the quality of our manuscript. We have included detailed responses to his comments and suggestions in the supplementary .pdf file. We have also included a modified version of the original manuscript. Please note the supplement to this comment and the modified manuscript, with modifications in blue.

**General comments:**

**Comment 1:**

*My most important point considers the calibration. It consists of 81 runs with three different values for the calibrated prefactors. I fully understand that running a distributed model has a high computation cost, but this number of runs seems rather low to me. There is a big risk of undersampling, leading to results that can just be a mere coincidence. For example, the most optimal value of $f_w$ may actually be 0.889, whereas only 0.75, 1 and 1.25 are explored in the study. Once again, I understand the burden of computational efforts, but at least the authors may want to reflect on this limitation in their discussion. In addition, it is mentioned that, except for these four prefactors, the remaining parameters were kept fixed. How many parameters are kept fixed and to what extend is the model already directed towards a certain solution by the choice of fixing certain parameters? For example, according to Figure 2 PCRGLOB-WB uses an interception routine. If the maximum interception capacity is kept fixed, it will probably influence the results for GLEAM versus the modelled evaporation.*

**Answer:**

We understand the reviewer's comment about the number of runs carried out for the study. We considered four different prefactors for calibration and we kept the remaining ones fixed (more information about these parameters can be found in Sutanudjaja et al., 2011). With this approach, we aimed to understand the state and uncertainty of model parameters, improving our knowledge about the importance or influence of each model parameter for each calibration scenario in this Moroccan basin. We agree with the reviewer that more runs (e.g. $f_w = \{0.75, 0.8, 0.85, \ldots, 1.25\}$) would be needed to precisely estimate the optimal parameters set. However, as the reviewer has pointed out, running a large-scale hydrological model, such as PCR-GLOBWB, requires high computational cost in terms of computer power and time. Therefore, we could not perform more runs at this stage. For future work, the range of prefactors values and then, the number of runs will be increased. Nevertheless, we will acknowledge this study limitation in section 5. Discussion and conclusions as follows:

P16L11: "… is selected. For these combinations, and due to computational limitations, only four prefactors were considered leading to 81 model runs per precipitation product. Using

more prefactor values and therefore, more runs may improve the estimation of the optimal parameters set for each calibration scenario. A suggestion …"

**Comment 2:**

*In addition to this, the step-wise calibration consists out of first calibrating on evaporation with GLEAM, and, in a second step, on soil moisture. I just wonder how much influence this order in calibration influences the results, especially as soil moisture strongly influences evaporation. Did you consider a step-wise calibration with first ESA CCI soil moisture and then GLEAM in a second step?*

**Answer:**

As the reviewer pointed out in his comment, the step-wise calibration scenario is based on both observations, evapotranspiration and soil moisture. In the first step, all prefactors are calibrated using GLEAM actual evapotranspiration and in the second step, those prefactors that have been identified are held constant and the remaining ones are calibrated according to ESA CCI soil moisture. Results showed that GLEAM actual evapotranspiration can be used to calibrate only the reference potential evapotranspiration ($f_e$), whereas ESA CCI soil moisture allows the identification of the minimum soil water capacity ($f_w$) and the saturated hydraulic conductivities of the soil layers ($f_k$). Therefore, in the step-wise calibration scenario firstly $f_e$ is identified and kept fixed, and secondly, $f_w$ and $f_k$ are identified. We agree with the reviewer that the calibration based on evapotranspiration also influences prefactors different to $f_e$ (and the calibration based on soil moisture also influences prefactors different to $f_w$ and $f_k$), but the impact is less significant and does not allow the identification of those prefactors. If we consider a step-wise calibration scenario using first ESA CCI soil moisture and GLEAM actual evapotranspiration in a second step, $f_w$ and $f_k$ would be first identified and maintain constant and $f_e$ would be identified later. This means that, for this particular study, a change in the order in which the observations are considered in the step-wise calibration scenario would not imply different results, because each observation allows the identification of different model prefactors.

**Comment 3:**

*I also wonder what the reasoning is behind the choice to compare the ESA CCI surface soil moisture with the soil moisture of the first three soil layers of PCR-GLOBWB. As mentioned by the authors (page 8, line 30) the ESA CCI soil moisture only represents the first 0.5-2cm, so wouldn't it make more sense to just compare with only the very first soil layer (first 5cm according to page 5, line 14) in PCR-GLOBWB? In this way, all parameters affecting the soil moisture in all the three layers will react, which can also be noted from the results for fk, but one could wonder whether this is for the right reason.*

**Answer:**

According to P7L27-28: "ESA CCI surface soil moisture observations were compared to simulated soil moisture with the first of the three vertical soil layers in PCR-GLOBWB." To clarify this and following the reviewer's suggestion, we will include a note as follows:

P7L23-24: "… ESA CCI surface soil moisture observations were compared to simulated soil moisture of the first of the three vertical soil layers in PCR-GLOBWB (top 5 cm of soil). …"

**Comment 4:**

*Often, a comparison is made between the reference scenario S0 and the scenarios S2- S4. Nevertheless, S0 is merely an uncalibrated model and especially for Ait Ouchene (Figure 6), the model performances are rather poor. Therefore, not much is needed to achieve improvements in this case. Isn't it much more interesting to focus more on comparing S1 with S2-S4? In other words, how close can we get to a calibration on streamflow with help of GLEAM and ESA CCI? It would be interesting to see if differences occur in Figures 8-10 for S1 and S2-4. Ideally, there would be no difference, but I expect that this will not be the case.*

**Answer:**

We agree with the reviewer and we will replace Figures 8, 9 and 10 with new figures (Figures 7 and 8). Figure 7 will show comparisons between estimated and observed evapotranspiration, soil moisture and discharge including calibration scenarios S0 (reference scenario) and S4 (step-wise calibration). Similarly, to Figure 7, Figure 8, will show comparisons between estimated and observed evapotranspiration, soil moisture and discharge including calibration scenarios S1 (in-situ discharge calibration) and S4 (step-wise calibration). These two figures should improve the presentation of the calibration results. At the same time, section 4, results will be restructure in two subsections: 4.1. Calibration results and 4.2. Validation results. Section 4.1. Calibration results will be adjusted accordingly (please see modifications in blue in the manuscript).

**Detailed comments (P: page, L: line or lines):**

**Comment 5:**

*P8.L2-3. I don't know if these specific stations were used for MSWEP, but as MSWEP used station data as input (also remarked by the authors on P7.L19-20), isn't it logical that MSWEP provided a better fit to the station data?*

**Answer:**

MSWEP, WFDEI and EI precipitation values were interpolated to two weather station locations to calculate various performance metrics. These weather stations were not used to generate any of the global precipitation products, ensuring an independent validation. Furthermore, this precipitation evaluation was carried out only for two rainfall stations that were found inside the Oum Er Rbia basin. These measurements were considered too scarce in number and spatially sparse to cover the entire basin and therefore to extract firm conclusions about which products outperforms the other ones.

**Comment 6:**

*P10.L15-17. Why model at a daily basis and only compare on monthly values? What is the temporal resolution of the data (discharge, GLEAM and ESA CCI)?*

**Answer:**

As the reviewer has mentioned, PCR-GLOBWB runs at a daily temporal resolution and the meteorological data, the GLEAM actual evapotranspiration and the ESA CCI soil moisture

observations are daily too. Preliminary analysis and comparison of the calibration scenarios were made at a daily temporal resolution initially. Daily and monthly results were similar. Therefore, for practical reasons and to simplify the manuscript, we decided to include results relative only to the monthly temporal resolution. Moreover, for water resources applications, monthly estimates can be sufficient in this particular area.

**Comment 7:**

*P12.L10. I can see that fw shows a clear pattern, but I don't see this clearly for fe.*

**Answer:**

We agree with the reviewer that $f_w$ can be well identified from Figures 5a and Figure 5b. The pattern for $f_e$ identification, although is not as clear as $f_w$, is also visible, especially in Figure 5b when WFDEI and MSWEP precipitation are used. Higher values of $f_e$, in particular $f_e = 1.25$, result in higher KGE values.

We will improve Figure 5 in different ways: we will use different colours and dot shapes to indicate different values of $f_e$, we will modify the horizontal axis of each scatterplot limiting the tick marks and numbers to the values of the used calibration prefactors and we will change the label of y-axis to indicate when KGE values are based on discharge, actual evapotranspiration and surface soil moisture using subscripts $KGE_q$, $KGE_{evap}$ and $KGE_{sm}$. Moreover, we will modify the figure analysis as follows:

P11L26-30: "… Figures 5a and 5b (calibration scenario S1) are similar. From these figures, $f_e$ (1$^{st}$ column) and $f_w$ (4$^{th}$ column) are well identified by discharge calibration at both gauging stations when forced with any of the three precipitation products. $f_e = 1.25$ and $f_w = 1.25$ lead to the highest $KGE_q$ values. However, it is not possible to identify the best prefactors of $f_j$ (2$^{nd}$ column) and $f_k$ (3$^{rd}$ column). There are no clear and distinct maximum values in the scatterplots of these figures, hence $f_j = 0$ and $f_k = 0$ are used. …"

**Comment 8:**

*P13.L16-17. These numbers refer to the WFDEI-case*

**Answer:**

We will correct these KGE values as follows:

P13L20-25: "… From Figure 6 (5$^{th}$ column), calibration using GLEAM evapotranspiration and ESA CCI soil moisture leads to further improvements than when these observations are separately used. For example, when MSWEP precipitation is used to model discharge at Mechra Eddahk station, $KGE_q$ varies between 0.703, 0.693, 0.613 and 0.573 for calibration scenarios S1, S4, S2 and S3, respectively ($KGE_q = 0.561$ for the reference scenario S0). At Ait Ouchene station (see Figure 2 in the Supplementary Information), $KGE_q$ varies between 0.520, 0.342, 0.331 and 0.271 for calibration scenarios S1, S4, S2 and S3, respectively ($KGE_q = 0.542$ for the reference scenario S0). …"

**Comment 9:**

*P14.L9-11. It may as well be model structural deficiencies as wrong parameterizations. It is a bit easy to blame the input data directly, especially as it happens for two out of three input*

*products. It must be noted as well that even though EI has the peaks in 2002 right, it also underestimates the peaks in 1996 and 1997.*

**Answer:**

Indeed, these differences may be also related with model structural deficiencies. We will modify the manuscript according to the reviewer's suggestion:

P14L13-17: "… scenarios. From Figure 7c, the step-wise calibrated run (S4) reproduces the monthly observed discharge well, except some simulated extreme peaks which were not observed, e.g. January and June in 2002 and some which were not simulated properly, e. g. January and May in 1996 and 1997. This lack of fit may be due to errors in the precipitation data, because higher discharge differences are shown when WFDEI and MSWEP products are used in comparison to EI product. Other possible reasons may be related with model structural deficiencies. When …"

**Comment 10:**

*Throughout the manuscript, the terms KGE, NSE etc. are used and sometimes refer to a case with evaporation and sometimes to cases with soil moisture or discharge. For clarity, it might be good to add a subscript (e.g. $KGE_E$, $KGE_{SM}$ etc.).*

**Answer:**

We believe that the reviewer's suggestion will facilitate the understanding of the manuscript, especially in section 4. Results. Therefore, we will modify the manuscript indicating with a subscript when the performance metrics are calculated for precipitation (precip), evapotranspiration (evap), soil moisture (sm) and discharge (q). Moreover, we will include the following sentence:

P10L8-10: "… is 0.

When the performance metrics were calculated between simulated and observed soil moisture estimates, the subscript sm was added to the metric, i.e. $NSE_{sm}$, $KGE_{sm}$, $RMSE_{sm}$, $MAE_{sm}$, $r_{sm}$ and $PBias_{sm}$. Similarly, when comparing actual evapotranspiration estimates, precipitation and discharge, the added subscripts were evap, precip and q, respectively. …"

**References:**

- Sutanudjaja, E. H., Van Beek, L. P. H., De Jong, S. M., van Geer, F. C., & Bierkens, M. F. P. (2011). Large-scale groundwater modeling using global datasets: a test case for the Rhine-Meuse basin. Hydrology and Earth System Sciences, 15(9), 2913.

**Additional modifications in figures and figures to be included**

[Figure]

***Figure 5.*** *Scatterplots of discharge performance indicator KGE based on the monthly observations versus prefactors $f_e$, $f_j$, $f_k$ and $f_w$ for the calibration scenarios S1 ((a) Ait Ouchene (b) Mechra Eddahk), S2 (c) and S3 (d). In each sub-figure, columns indicate the different calibrated prefactors and rows indicate the three global precipitation products used as model forcing. Different colours and dot shapes indicate different $f_w$ values.*

[Figure]

***Figure 7.*** *(a) Monthly GLEAM actual evapotranspiration (black) and estimated actual evapotranspiration (red and purple) time series. (b) Monthly ESA CCI soil moisture (black) and estimated soil moisture (red and purple) time series. (c) Monthly observed discharge (black) and estimated discharge (red and purple) time series. The red dashed lines represent estimates from calibration scenario S0 (reference scenario). The purple dashed lines represent the calibrated time series from calibration scenario S4 which are taken from the runs that yield the best simulations. Estimated time series over the entire Oum Er Rbia basin for the validation time period obtained with MSWEP precipitation are shown.*

[Figure]

*Figure 8.* (a) Monthly GLEAM actual evapotranspiration (black) and estimated actual evapotranspiration (red and purple) time series. (b) Monthly ESA CCI soil moisture (black) and estimated soil moisture (red and purple) time series. (c) Monthly observed discharge (black) and estimated discharge (red and purple) time series. The red dashed lines represent estimates from calibration scenario S1. The purple dashed lines represent the calibrated time series from calibration scenario S4 which are taken from the runs that yield the best simulations. Estimated time series over the entire Oum Er Rbia basin for the validation time period obtained with MSWEP precipitation are shown.

---

## Author Comment (AC4) · 21 Apr 2017

The comment was uploaded in the form of a supplement:
http://www.hydrol-earth-syst-sci-discuss.net/hess-2017-16/hess-2017-16-AC4-supplement.pdf

---

## Author Response (AR1)

**Response to Prof. Markus Hrachowitz on the manuscript "Calibration of a large-scale hydrological model using satellite-based soil moisture and evapotranspiration products"**

**Patricia López López, Edwin H. Sutanudjaja, Jaap Schellekens, Geert Sterk and Marc F. P. Bierkens**

The authors thank Prof. Markus Hrachowitz for his comments and recommendations on the manuscript. According to them, we have focused the modifications of the manuscript on its length and the presentation of the results. We have also modified the introduction and discussion and conclusions sections to better structure the manuscript and to clarify the purpose of the research study. A new version of the manuscript is included, where modifications are highlighted in blue.

---

## Author Response (AR2)

**Response to Dr. Heye R. Bogena on the manuscript**
**"Calibration of a large-scale hydrological model using satellite-based soil moisture and evapotranspiration products"**
**Patricia López López, Edwin H. Sutanudjaja, Jaap Schellekens, Geert Sterk and Marc F. P. Bierkens**

The authors would like to thank the reviewer for his technical comments on the manuscript. We have addressed them and modified the manuscript accordingly.

**Comment 1:**

*P2 L13-15: The format of this enumeration is inappropriate. Please reformulate.*

**Answer:**

We will modify the enumeration of examples of hydro-meteorological datasets t as follows:

P2L13-15:"… hydro-meteorological datasets at finer spatial and temporal resolutions: precipitation (Joyce et al., 2004; Huffman et al., 2007), soil moisture (Njoku et al., 2003; Dorigo et al., 2015), total water storage (Tapley et al., 2004), evapotranspiration (Bastiaanssen et al., 1998; Nishida, 2003; Miralles et al., 2011b), etc. …"

**Comment 2:**

*P2 L18-19: Inappropriate use of hyphens.*

**Answer:**

We will delete the hyphens and modify the manuscript as follows:

P2L18-19:"… and to improve streamflow model estimates through assimilation (Parajka et al., 2006; Roy et al., 2010; Brocca et al., 2012; Thirel et al., 2013; López López et al., 2016) and/or calibration techniques or a-priori determination of model parameters (Jacobs et al., 2003; Beck et al., 2009). Calibration. …"

**Comments 3 and 4:**

*P3 L14-15: Web links should not be presented within the main text. Please properly cite within the reference list and indicate the last date of access.*

*P4 L23: Web links should not be presented within the main text. Please properly cite within the reference list and indicate the last date of access.*

**Answer:**

We will delete the web links in the manuscript and we will include the correct reference in References section.

**Comment 5:**

*P17 L12: The sentence is incomplete. Please reformulate.*

**Answer:**

We will reformulate the sentence and we will also correct the enumeration of the lines in the manuscript.